# Adversarial Model for Offline Reinforcement Learning

**Mohak Bhardwaj**[*]
University of Washington
mohakb@cs.washington.edu

**Tengyang Xie**[*]
Microsoft Research & UW-Madison
tx@cs.wisc.edu

**Byron Boots**
University of Washington
bboots@cs.washington.edu

**Nan Jiang**
UIUC
nanjiang@illinois.edu

**Ching-An Cheng**
Microsoft Research, Redmond
chinganc@microsoft.com

## Abstract

We propose a novel model-based offline Reinforcement Learning (RL) framework, called Adversarial Model for Offline Reinforcement Learning (ARMOR), which can robustly learn policies to improve upon an arbitrary reference policy regardless of data coverage. ARMOR is designed to optimize policies for the worst-case performance relative to the reference policy through adversarially training a Markov decision process model. In theory, we prove that ARMOR, with a well-tuned hyperparameter, can compete with the best policy within data coverage when the reference policy is supported by the data. At the same time, ARMOR is robust to hyperparameter choices: the policy learned by ARMOR, with *any* admissible hyperparameter, would never degrade the performance of the reference policy, even when the reference policy is not covered by the dataset. To validate these properties in practice, we design a scalable implementation of ARMOR, which by adversarial training, can optimize policies without using model ensembles in contrast to typical model-based methods. We show that ARMOR achieves competent performance with both state-of-the-art offline model-free and model-based RL algorithms and can robustly improve the reference policy over various hyperparameter choices.[2]

## 1   Introduction

Offline reinforcement learning (RL) is a technique for learning decision-making policies from logged data (Lange et al., 2012; Levine et al., 2020; Jin et al., 2021; Xie et al., 2021a). In comparison with alternate learning techniques, such as off-policy RL and imitation learning (IL), offline RL reduces the data assumption needed to learn good policies and does not require collecting new data. Theoretically, offline RL can learn the best policy that the given data can explain: as long as the offline data includes the scenarios encountered by a near-optimal policy, an offline RL algorithm can learn such a near-optimal policy, even when the data is collected by highly sub-optimal policies and/or is not diverse. Such robustness to data coverage makes offline RL a promising technique for solving real-world problems, as collecting diverse or expert-quality data in practice is often expensive or simply infeasible.

The fundamental principle behind offline RL is the concept of pessimism, which considers worst-case outcomes for scenarios without data. In algorithms, this is realized by (explicitly or implicitly) constructing performance lower bounds in policy learning which penalizes uncertain actions.

---

[*]Equal contribution
[2]Open source code is available at: https://sites.google.com/view/armorofflinerl/.

37th Conference on Neural Information Processing Systems (NeurIPS 2023).

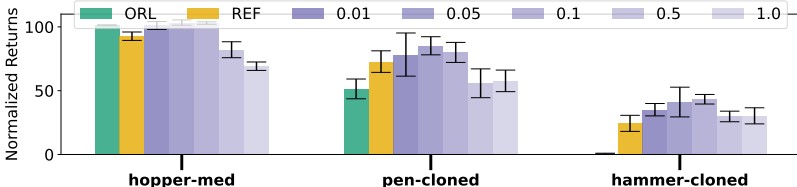

Figure 1: Robust Policy Improvement: ARMOR can improve performance over the reference policy (REF) over a broad range of pessimism hyperparameter (purple) regardless of data coverage. ORL denotes best offline RL policy without using the reference policy, and reference is obtained by behavior cloning on expert dataset.

Various designs have been proposed to construct such lower bounds, including behavior regularization (Fujimoto et al., 2019; Kumar et al., 2019; Wu et al., 2019; Laroche et al., 2019; Fujimoto and Gu, 2021), point-wise pessimism based on negative bonuses or truncation (Kidambi et al., 2020; Jin et al., 2021), value penalty (Kumar et al., 2020; Yu et al., 2020), or two-player games (Xie et al., 2021a; Uehara and Sun, 2021; Cheng et al., 2022). Conceptually, the tighter the lower bound is, the better the learned policy would perform; see a detailed discussion of related work in Appendix C.

Despite these advances, offline RL still has not been widely adopted to build learning-based decision systems beyond academic research. One important factor we posit is the issue of performance degradation: Usually, the systems we apply RL to have currently running policies, such as an engineered autonomous driving rule or a heuristic-based system for diagnosis, and the goal of applying a learning algorithm is often to further improve upon these baseline *reference policies*. As a result, it is imperative that the policy learned by the algorithm does not degrade the base performance. This criterion is especially critical for applications where poor decision outcomes cannot be tolerated.

However, running an offline RL algorithm based on pessimism, in general, is not free from performance degradation. While there have been algorithms with policy improvement guarantees (Laroche et al., 2019; Fujimoto et al., 2019; Kumar et al., 2020; Fujimoto and Gu, 2021; Cheng et al., 2022), such guarantees apply only to the behavior policy that collects the data, which might not necessarily be the reference policy. In fact, quite often these two policies are different. For example, in robotic manipulation, it is common to have a dataset of activities different from the target task. In such a scenario, comparing against the behavior policy is meaningless, as these policies do not have meaningful performance in the target task.

In this work, we propose a novel model-based offline RL framework, called Advesarial Model for Offline Rinforcement Learning (ARMOR), which can robustly learn policies that improve upon an arbitrary reference policy by adversarially training a Markov decision process (MDP) model, regardless of the data quality. ARMOR is designed based on the concept of relative pessimism (Cheng et al., 2022), which aims to optimize for the worst-case relative performance over uncertainty. In theory, we prove that, owing to relative pessimism, the ARMOR policy never degrades the performance of the reference policy for a range of hyperparameters which is given beforehand, a property known as Robust Policy Improvement (RPI) (Cheng et al., 2022). In addition, when the right hyperparameter is chosen, and the reference policy is covered by the data, we prove that the ARMOR policy can also compete with any policy covered by the data in an absolute sense. To our knowledge, RPI property of offline RL has so far been limited to comparing against the data collection policy (Fujimoto et al., 2019; Kumar et al., 2019; Wu et al., 2019; Laroche et al., 2019; Fujimoto and Gu, 2021; Cheng et al., 2022). In ARMOR, by adversarially training an MDP model, we extend the technique of relative pessimism to achieve RPI with *arbitrary* reference policies, regardless of whether they collected the data or not (Fig. 1).

In addition to theory, we design a scalable deep-learning implementation of ARMOR to validate these claims that jointly trains an MDP model and the state-action value function to minimize the estimated performance difference between the policy and the reference using model-based rollouts. Our implementation achieves state-of-the-art (SoTA) performance on D4RL benchmarks (Fu et al., 2020), while using only a *single* model (in contrast to ensembles used in existing model-based offline RL works). This makes ARMOR a better framework for using high-capacity world models (e.g.(Hafner et al., 2023)) for which building an ensemble is too expensive. We also empirically validate the RPI property of our implementation.

## 2 Preliminaries

**Markov Decision Process**   We consider learning in the setup of an infinite-horizon discounted Markov Decision Process (MDP). An MDP $M$ is defined by the tuple $\langle \mathcal{S}, \mathcal{A}, P_M, R_M, \gamma \rangle$, where $\mathcal{S}$ is the state space, $\mathcal{A}$ is the action space, $P_M : \mathcal{S} \times \mathcal{A} \to \Delta(\mathcal{S})$ is the transition dynamics, $R_M : \mathcal{S} \times \mathcal{A} \to [0,1]$ is a scalar reward function and $\gamma \in [0,1)$ is the discount factor. A policy $\pi$ is a mapping from $\mathcal{S}$ to a distribution on $\mathcal{A}$. For $\pi$, we let $d_M^\pi(s,a)$ denote the discounted state-action distribution obtained by running $\pi$ on $M$ from an initial state distribution $d_0$, i.e $d_M^\pi(s,a) = (1-\gamma)\mathbb{E}_{\pi,M}\left[\sum_{t=0}^\infty \gamma^t \mathbb{1}(s_t = s, a_t = a)\right]$. Let $J_M(\pi) = \mathbb{E}_{\pi,M}\left[\sum_{t=0}^\infty \gamma^t r_t\right]$ be the expected discounted return of policy $\pi$ on $M$ starting from $d_0$, where $r_t = R_M(s_t, a_t)$. We define the value function as $V_M^\pi(s) = \mathbb{E}_{\pi,M}\left[\sum_{t=0}^\infty \gamma^t r_t | s_0 = s\right]$, and the state-action value function (i.e., Q-function) as $Q_M^\pi(s,a) = \mathbb{E}_{\pi,M}\left[\sum_{t=0}^\infty \gamma^t r_t | s_0 = s, s_0 = a\right]$. By this definition, we note $J_M(\pi) = \mathbb{E}_{d_0}[V_M^\pi(s)] = \mathbb{E}_{d_0,\pi}[Q_M^\pi(s,a)]$. We use $[0, V_{\max}]$ to denote the range of value functions, where $V_{\max} \geq 1$. We denote the ground truth MDP as $M^\star$, and $J = J_{M^\star}$

**Offline RL**   The aim of offline RL is to find the policy that maximizes $J(\pi)$, while using a fixed dataset $\mathcal{D}$ collected by a behavior policy $\mu$. We assume the dataset $\mathcal{D}$ consists of $\{(s_n, a_n, r_n, s_{n+1})\}_{n=1}^N$, where $(s_n, a_n)$ is sampled from $d_{M^\star}^\mu$ and $r_n, s_{n+1}$ follow $M^\star$; for simplicity, we also write $\mu(s,a) = d_{M^\star}^\mu(s,a)$.

We assume that the learner has access to a Markovian policy class $\Pi$ and an MDP model class $\mathcal{M}$.

**Assumption 1** (Realizability). *We assume the ground truth model $M^\star$ is in the model class $\mathcal{M}$.*

In addition, we assume that we are provided a reference policy $\pi_{\mathsf{ref}}$. In practice, such a reference policy represents a baseline whose performance we want to improve with offline RL and data.

**Assumption 2** (Reference policy). *We assume access to a reference policy $\pi_{\mathsf{ref}}$, which can be queried at any state. We assume $\pi_{\mathsf{ref}}$ is realizable, i.e., $\pi_{\mathsf{ref}} \in \Pi$.*

If $\pi_{\mathsf{ref}}$ is not provided, we can still run ARMOR as a typical offline RL algorithm, by first performing behavior cloning on the data and setting the cloned policy as $\pi_{\mathsf{ref}}$. In this case, ARMOR has RPI with respect to the behavior policy.

**Robust Policy Improvement**   RPI is a notion introduced in Cheng et al. (2022), which means that the offline algorithm can learn to improve over the behavior policy, using hyperparameters within a known set. Algorithms with RPI are more robust to hyperparameter choices, and they are often derived from the principle of relative pessimism (Cheng et al., 2022). In this work, we extend the RPI concept to compare with an arbitrary reference (or baseline) policy, which can be different from the behavior policy and can take actions outside data support.

## 3 Adversarial Model for Offline Reinforcement Learning (ARMOR)

ARMOR is a model-based offline RL algorithm designed with relative pessimism. The goal of ARMOR is to find a policy $\widehat{\pi}$ that maximizes the performance difference $J(\widehat{\pi}) - J(\pi_{\mathsf{ref}})$ to a given reference policy $\pi_{\mathsf{ref}}$, while accounting for the uncertainty due to limited data coverage. ARMOR achieves this by solving a two-player game between a learner policy and an adversary MDP model:

$$\widehat{\pi} = \underset{\pi \in \Pi}{\operatorname{argmax}}\ \min_{M \in \mathcal{M}_\alpha} J_M(\pi) - J_M(\pi_{\mathsf{ref}}) \tag{1}$$

based on a version space of MDP models

$$\mathcal{M}_\alpha = \{M \in \mathcal{M} : \mathcal{E}_\mathcal{D}(M) - \min_{M' \in \mathcal{M}} \mathcal{E}_\mathcal{D}(M') \leq \alpha\}, \tag{2}$$

where we define the model fitting loss as

$$\mathcal{E}_\mathcal{D}(M) \coloneqq -\sum_\mathcal{D} \log P_M(s' \mid s, a) + {}^{(R_M(s,a)-r)^2}/V_{\max}^2 \tag{3}$$

and $\alpha \geq 0$ is a bound on statistical errors such that $M^\star \in \mathcal{M}_\alpha$. In this two-player game, ARMOR is optimizing a lower bound of the relative performance $J(\pi) - J(\pi_{\mathsf{ref}})$. This is due to the construction that $M^\star \in \mathcal{M}_\alpha$, which ensures $\min_{M \in \mathcal{M}_\alpha} J_M(\pi) - J_M(\pi_{\mathsf{ref}}) \leq J_{M^\star}(\pi) - J_{M^\star}(\pi_{\mathsf{ref}})$.

One interesting property that follows from optimizing the relative performance lower bound is that $\widehat{\pi}$ is guaranteed to always be no worse than $\pi_{\mathsf{ref}}$, for a wide range of $\alpha$ and regardless of the relationship between $\pi_{\mathsf{ref}}$ and the data $\mathcal{D}$.

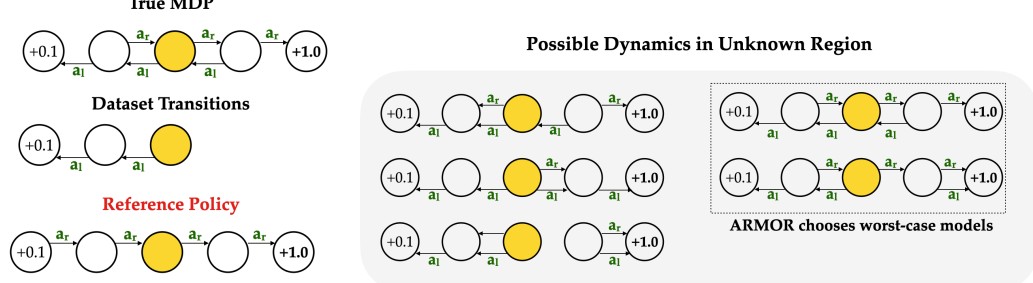

Figure 2: A toy MDP illustrating the RPI property of ARMOR. (Top) The true MDP has deterministic dynamics where taking the left ($a_l$) or right ($a_r$) actions takes the agent to corresponding states; start state is in yellow. The suboptimal behavior policy visits only the left part of the state space, and the reference policy demonstrates optimal behavior by always choosing $a_r$. (Bottom) A subset of possible data-consistent MDP models in the version space. The adversary always chooses the MDP that makes the reference maximally outperform the learner. In response, the learner will learn to mimic the reference outside data support to be competitive.

**Proposition 1.** *For any $\alpha$ large enough such that $M^\star \in \mathcal{M}_\alpha$, it holds that $J(\widehat{\pi}) \geq J(\pi_{\mathsf{ref}})$.*

This fact can be easily reasoned: Since $\pi_{\mathsf{ref}} \in \Pi$, we have $\max_{\pi \in \Pi} \min_{M \in \mathcal{M}_\alpha} J_M(\pi) - J_M(\pi_{\mathsf{ref}}) \geq \min_{M \in \mathcal{M}_\alpha} J_M(\pi_{\mathsf{ref}}) - J_M(\pi_{\mathsf{ref}}) = 0$. In other words, ARMOR achieves the RPI property with respect to any reference policy $\pi_{\mathsf{ref}}$ and offline dataset $\mathcal{D}$.

This RPI property of ARMOR is stronger than the RPI property in the literature. In comparison, previous algorithms with RPI (Fujimoto et al., 2019; Kumar et al., 2019; Wu et al., 2019; Laroche et al., 2019; Fujimoto and Gu, 2021; Cheng et al., 2022) are only guaranteed to be no worse than the behavior policy that collected the data. In Section 3.2, we will also show that when $\alpha$ is set appropriately, ARMOR can provably compete with the best data covered policy as well, as prior offline RL works (e.g., Xie et al., 2021a; Uehara and Sun, 2021; Cheng et al., 2022).

## 3.1 An Illustrative Toy Example

Why does ARMOR have the RPI property, even when the reference policy $\pi_{\mathsf{ref}}$ is not covered by the data $\mathcal{D}$? While we will give a formal analysis soon in Section 3.2, here we provide some intuitions as to why this is possible. First, notice that ARMOR has access to the reference policy $\pi_{\mathsf{ref}}$. Therefore, a trivial way to achieve RPI with respect to $\pi_{\mathsf{ref}}$ is to just output $\pi_{\mathsf{ref}}$. However, this naïve algorithm while never degrading $\pi_{\mathsf{ref}}$ cannot learn to improve from $\pi_{\mathsf{ref}}$. ARMOR achieves these two features simultaneously by *1)* learning an MDP Model, and *2)* adversarially training this MDP model to minimize the relative performance difference to $\pi_{\mathsf{ref}}$ during policy optimization.

We illustrate this by a one-dimensional discrete MDP example with five possible states as shown in Figure 2. The dynamic is deterministic, and the agent always starts in the center cell. The agent receives a lower reward of 0.1 in the left-most state and a high reward of 1.0 upon visiting the right-most state. Say, the agent only has access to a dataset from a sub-optimal policy that always takes the left action to receive the 0.1 reward. Further, let's say we have access to a reference policy that demonstrates optimal behavior on the true MDP by always visiting the right-most state. However, it is unknown a priori that the reference policy is optimal. In such a case, typical offline RL methods can only recover the sub-optimal policy from the dataset as it is the best-covered policy in the data.

ARMOR can learn to recover the expert reference policy in this example by performing rollouts with the adversarially trained MDP model. From the realizability assumption (Assumption 1), we know that the version space of models contains the true model (i.e., $M^\star \in \mathcal{M}_\alpha$). The adversary can then choose a model from this version space where the reference policy $\pi_{\mathsf{ref}}$ maximally outperforms the learner. In this toy example, the model selected by the adversary would be the one allowing the expert policy to reach the right-most state. Now, optimizing relative performance difference with respect to this model will ensure that the learner can recover the expert behavior, since the only way for the learner to stay competitive with the reference policy is to mimic the reference policy in the region outside data support. In other words, the reason why ARMOR has RPI to $\pi_{\mathsf{ref}}$ is that

its adversarial model training procedure can augment the original offline data with new states and actions that would cover those generated by running the reference policy.[3]

## 3.2 Theoretical Analysis

Now we make the above discussions formal and give theoretical guarantees on ARMOR's absolute performance and RPI property. To this end, we introduce a single-policy concentrability coefficient, which measures the distribution shift between a policy $\pi$ and the data distribution $\mu$.

**Definition 1** (Generalized Single-policy Concentrability). *We define the generalized single-policy concentrability for policy $\pi$, model class $\mathcal{M}$ and offline data distribution $\mu$ as* $\mathfrak{C}_{\mathcal{M}}(\pi) := \sup_{M \in \mathcal{M}} \frac{\mathbb{E}_{d^{\pi}}[\mathcal{E}^{\star}(M)]}{\mathbb{E}_{\mu}[\mathcal{E}^{\star}(M)]}$, *where* $\mathcal{E}^{\star}(M) = D_{\mathrm{TV}}\left(P_M(\cdot \mid s, a), P_{M^{\star}}(\cdot \mid s, a)\right)^2 + (R_M(s,a) - R^{\star}(s,a))^2/V_{\max}^2$.

Note that $\mathfrak{C}_{\mathcal{M}}(\pi)$ is always upper bounded by the standard single-policy concentrability coefficient $\|d^{\pi}/\mu\|_{\infty}$ (e.g., Jin et al., 2021; Rashidinejad et al., 2021; Xie et al., 2021b), but it can be smaller in general with model class $\mathcal{M}$. It can also be viewed as a model-based analog of the one in Xie et al. (2021a). A detailed discussion around $\mathfrak{C}_{\mathcal{M}}(\pi)$ can be found in Uehara and Sun (2021).

First, we present the absolute performance guarantee of ARMOR, which holds for a well-tuned $\alpha$.

**Theorem 2** (Absolute performance). *Under Assumption 1, there is an absolute constant $c$ such that for any $\delta \in (0, 1]$, if we set $\alpha = c \cdot (\log(|\mathcal{M}|/\delta))$ in Eq. (2), then for any reference policy $\pi_{\mathrm{ref}}$ and comparator policy $\pi^{\dagger} \in \Pi$, with probability $1 - \delta$, the policy $\widehat{\pi}$ learned by ARMOR in Eq. (1) satisfies that $J(\pi^{\dagger}) - J(\widehat{\pi})$ is upper bounded by*

$$\mathcal{O}\left(\left(\sqrt{\mathfrak{C}_{\mathcal{M}}(\pi^{\dagger})} + \sqrt{\mathfrak{C}_{\mathcal{M}}(\pi_{\mathrm{ref}})}\right) \frac{V_{\max}}{1-\gamma} \sqrt{\frac{\log(|\mathcal{M}|/\delta)}{n}}\right).$$

Roughly speaking, Theorem 2 shows that $\widehat{\pi}$ learned by ARMOR can compete with any policy $\pi^{\dagger}$ with a large enough dataset, as long as the offline data $\mu$ has good coverage on $\pi^{\dagger}$ (good coverage over $\pi_{\mathrm{ref}}$ can be automatically satisfied if we simply choose $\pi_{\mathrm{ref}} = \mu$, which yields $\mathfrak{C}_{\mathcal{M}}(\pi_{\mathrm{ref}}) = 1$). Compared to the closest model-based offline RL work (Uehara and Sun, 2021), if we set $\pi_{\mathrm{ref}} = \mu$ (data collection policy), Theorem 2 leads to almost the same guarantee as Uehara and Sun (2021, Theorem 1) up to constant factors.

In addition to absolute performance, below we show that, under Assumptions 1 and 2, ARMOR has the RPI property to $\pi_{\mathrm{ref}}$: it always improves over $J(\pi_{\mathrm{ref}})$ for *a wide range of parameter $\alpha$*. Compared with the model-free ATAC algorithm in Cheng et al. (2022, Proposition 6), the threshold for $\alpha$ in Theorem 3 does not depend on sample size $N$ due to the model-based nature of ARMOR.

**Theorem 3** (Robust strong policy improvement). *Under Assumptions 1 and 2, there exists an absolute constant $c$ such that for any $\delta \in (0, 1]$, if: i) $\alpha \geq c \cdot (\log(|\mathcal{M}|/\delta))$ in Eq. (2); ii) $\pi_{\mathrm{ref}} \in \Pi$, then with probability $1 - \delta$, the policy $\widehat{\pi}$ learned by ARMOR in Eq. (1) satisfies $J(\widehat{\pi}) \geq J(\pi_{\mathrm{ref}})$.*

The detailed proofs of Theorems 2 and 3, as well as the discussion on how to relax Assumptions 1 and 2 to the misspecified model and policy classes are deferred to Appendix A.

# 4 Practical Implementation

In this section, we present a scalable implementation of ARMOR (Algorithm 1) that approximately solves the two-player game in Eq. (1). We first describe the overall design principle and then the algorithmic details.

## 4.1 A Model-based Actor Critic Approach

For computational efficiency, we take a model-based actor critic approach and solve a regularized version of Eq. (1). We construct this regularized version by relaxing the constraint $M \in \mathcal{M}_{\alpha}$

---

[3]Note that ARMOR does not depend on knowledge of the true reward function and similar arguments hold in the case of learned rewards as we illustrate in Appendix E.

**Algorithm 1** ARMOR (Adversarial Model for Offline Reinforcement Learning)

---

**Input:** Batch data $\mathcal{D}_{\text{real}}$, policy $\pi$, MDP model $M$, critics $f_1, f_2$, horizon $H$, constants $\beta, \lambda \geq 0$, $\tau \in [0, 1]$, $w \in [0, 1]$,

1: Initialize target networks $\bar{f}_1 \leftarrow f_1, \bar{f}_2 \leftarrow f_2$ and $\mathcal{D}_{\text{model}} = \emptyset$
2: **for** $k = 0, \ldots, K - 1$ **do**
3:     Sample minibatch $\mathcal{D}_{\text{real}}^{\text{mini}}$ from dataset $\mathcal{D}_{\text{real}}$ and minibatch $\mathcal{D}_{\text{model}}^{\text{mini}}$ from dataset $\mathcal{D}_{\text{model}}$.
4:     Construct transition tuples using model predictions

$$\mathcal{D}_M := \left\{ (s, a, r_M, s'_M) : r_M = R_M(s, a), s'_M \sim P_M(\cdot \mid s, a), (s, a) \in \mathcal{D}_{\text{real}}^{\text{mini}} \cup \mathcal{D}_{\text{model}}^{\text{mini}} \right\}$$

5:     Update the adversary networks; for $i = 1, 2$,

$$l^{\text{adversary}}(f, M) := \mathcal{L}_{\mathcal{D}_M}(f, \pi, \pi_{\text{ref}}) + \beta \left( \mathcal{E}_{\mathcal{D}_M}^w (f, M, \pi) + \lambda \mathcal{E}_{\mathcal{D}_{\text{real}}^{\text{mini}}}(M) \right) \tag{4}$$

$$M \leftarrow M - \eta_{\text{fast}} \left( \nabla_M l^{\text{adversary}}(f_1, M) + \nabla_M l^{\text{adversary}}(f_2, M) \right)$$

$$f_i \leftarrow \text{Proj}_{\mathcal{F}}(f_i - \eta_{\text{fast}} \nabla_{f_i} l^{\text{adversary}}(f_i, M)) \quad \text{and} \quad \bar{f}_i \leftarrow (1 - \tau) \bar{f}_i + \tau f_i$$

6:     Update actor network with respect to the first critic network and the reference policy

$$l^{\text{actor}}(\pi) := -\mathcal{L}_{\mathcal{D}_M}(f_1, \pi, \pi_{\text{ref}}) \tag{5}$$

$$\pi \leftarrow \text{Proj}_{\Pi}(\pi - \eta_{\text{slow}} \nabla_{\pi} l^{\text{actor}}(\pi))$$

7:     If $k\%H = 0$, then reset model state: $\bar{S}_{\pi} \leftarrow \{s \in \mathcal{D}_{\text{real}}^{\text{mini}}\}$ and $\bar{S}_{\pi_{\text{ref}}} \leftarrow \{s \in \mathcal{D}_{\text{real}}^{\text{mini}}\}$
8:     Query the MDP model to expand $\mathcal{D}_{\text{model}}$ and update model state

$$\bar{A}_{\pi} := \{a : a \sim \pi(s), s \in \bar{S}_{\pi}\} \quad \text{and} \quad \bar{A}_{\pi_{\text{ref}}} := \{a : a \sim \pi_{\text{ref}}(s), s \in \bar{S}_{\pi_{\text{ref}}}\}$$

$$\mathcal{D}_{\text{model}} := \mathcal{D}_{\text{model}} \cup \{\bar{S}_{\pi}, \bar{A}_{\pi}\} \cup \{\bar{S}_{\pi_{\text{ref}}}, \bar{A}_{\pi_{\text{ref}}}\}$$

$$\bar{S}_{\pi} \leftarrow \{s' \mid s' \sim \texttt{detach}(P_M(\cdot \mid s, a)), s \in \bar{S}_{\pi}, a \in \bar{A}_{\pi}\}$$

$$\bar{S}_{\pi_{\text{ref}}} \leftarrow \{s' \mid s' \sim \texttt{detach}(P_M(\cdot \mid s, a)), s \in \bar{S}_{\pi_{\text{ref}}}, a \in \bar{A}_{\pi_{\text{ref}}}\}$$

9: **end for**

---

in the inner minimization of Eq. (1) to a regularization term and introducing an additional critic function. To clearly elaborate this, we first present the regularized objective in its complete form, and subsequently derive it from Eq. (1).

Let $\mathcal{F} : \{f : \mathcal{S} \times \mathcal{A} \to [0, V_{\max}]\}$ be a class of critic functions. The regularized objective is given as

$$\tilde{\pi} \in \underset{\pi \in \Pi}{\text{argmax}} \ \mathcal{L}_{d_M^{\pi_{\text{ref}}}}(\pi, f) \tag{6}$$

$$\text{s.t.} \ f^{\pi} \in \underset{M \in \mathcal{M}, f \in \mathcal{F}}{\text{argmin}} \ \mathcal{L}_{d_M^{\pi_{\text{ref}}}}(\pi, f) + \beta \left( \mathcal{E}_{\rho_{\pi_{\text{ref}}}, \pi}(\pi, f, M) + \lambda \mathcal{E}_{\mathcal{D}}(M) \right)$$

where $\mathcal{E}_{\mathcal{D}}(M) = \sum_{\mathcal{D}} -\log P_M(s' \mid s, a) + {(R_M(s,a)-r)^2}/{V_{\max}^2}$ is the model-fitting error, $\mathcal{L}_{d_M^{\pi_{\text{ref}}}}(\pi, f) := \mathbb{E}_{d_M^{\pi_{\text{ref}}}}[f(s, \pi) - f(s, \pi_{\text{ref}})]$ is equal to the performance difference $(1 - \gamma)(J_M(\pi) - J_M(\pi_{\text{ref}}))$, $\mathcal{E}_{\rho_{\pi_{\text{ref}}}, \pi}(\pi, f, M)$ denotes the squared Bellman error on the distribution $\rho_{\pi_{\text{ref}}, \pi}$ that denotes the distribution generated by first running $\pi_{\text{ref}}$ and then rolling out $\pi$ in $M$ (with a switching time sampled from a geometric distribution of $\gamma$), and $\beta, \lambda$ act as the Lagrange multipliers.

This regularized formulation in Eq. (6) can be derived as follows. Assuming $Q_M^{\pi} \in \mathcal{F}$, and using the facts that $J_M(\pi) = \mathbb{E}_{d_0}[Q_M^{\pi}(s, \pi)]$ and the Bellman equation $Q_M^{\pi}(s, a) = r_M(s, a) + \gamma \mathbb{E}_{s' \sim P_M(s,a)}[Q_M^{\pi}(s', \pi)]$, we can rewrite Eq. (1) as

$$\max_{\pi \in \Pi} \min_{M \in \mathcal{M}, f \in \mathcal{F}} \ \mathbb{E}_{d_M^{\pi_{\text{ref}}}}[f(s, \pi) - f(s, \pi_{\text{ref}})]. \tag{7}$$

$$\text{s.t.} \ \mathcal{E}_{\mathcal{D}}(M) \leq \alpha + \min_{M' \in \mathcal{M}} \mathcal{E}_{\mathcal{D}}(M')$$

$$\forall s, a \in \text{supp}(\rho_{\pi_{\text{ref}}, \pi}), \quad f(s, a) = r_M(s, a) + \gamma \mathbb{E}_{s' \sim P_M(s,a)}[f(s', \pi)]$$

We then convert the constraints in Eq. (7) into regularization terms in the inner minimization by introducing Lagrange multipliers ($\beta, \lambda$), following (Xie et al., 2021a; Cheng et al., 2022), and drop the constants not affected by $M, f, \pi$, which results in Eq. (6).

## 4.2 Algorithm Details

Algorithm 1 is an iterative solver for approximating the solution to Eq. (6). Here we further approximate $d_M^{\pi_{\mathsf{ref}}}$ and $\rho_{\pi_{\mathsf{ref}}, \pi}$ in Eq. (6) using samples from the state-action buffer $\mathcal{D}_{\mathsf{model}}$. We want ensure that $\mathcal{D}_{\mathsf{model}}$ has a larger coverage than both $d_M^{\pi_{\mathsf{ref}}}$ and $\rho_{\pi_{\mathsf{ref}}, \pi}$. We do so heuristically, by constructing the model replay buffer $\mathcal{D}_{\mathsf{model}}$ through repeatedly rolling out $\pi$ and $\pi_{\mathsf{ref}}$ with the adversarially trained MDP model $M$, such that $\mathcal{D}_{\mathsf{model}}$ contains a diverse training set of state-action tuples.

Specifically, the algorithm takes as input an offline dataset $\mathcal{D}_{\mathsf{real}}$, a policy $\pi$, an MDP model $M$ and two critic networks $f_1, f_2$. At every iteration, the algorithm proceeds in two stages. First, the adversary is optimized to find a data-consistent model that minimizes the performance difference with the reference policy. We sample mini-batches of only states and actions $\mathcal{D}_{\mathsf{real}}^{\mathsf{mini}}$ and $\mathcal{D}_{\mathsf{model}}^{\mathsf{mini}}$ from the real and model-generated datasets respectively (Line 4). The MDP model $M$ is queried on these mini-batches to generate next-state and reward predictions. The adversary then updates the model and Q-functions (Line 5) using the gradient of the loss described in Eq. (4), where

$$\mathcal{L}_{\mathcal{D}_M}(f, \pi, \pi_{\mathsf{ref}}) := \mathbb{E}_{\mathcal{D}_M}[f(s, \pi(s)) - f(s, \pi_{\mathsf{ref}}(s)]$$

$$\mathcal{E}_{\mathcal{D}_M}^w(f, M, \pi) := (1-w)\mathcal{E}_{\mathcal{D}}^{td}(f, f, M, \pi) + w\mathcal{E}_{\mathcal{D}}^{td}(f, \bar{f}, M, \pi)$$

$$\mathcal{E}_{\mathcal{D}_{\mathsf{real}}^{\mathsf{mini}}}(M) := \mathbb{E}_{\mathcal{D}_{\mathsf{real}}^{\mathsf{mini}}}[-\log P_M(s' \mid s, a) + {}^{(R_M(s,a)-r)^2}/V_{\max}^2]$$

$\mathcal{L}_{\mathcal{D}_M}$ is the pessimistic loss term that forces the $f$ to predict a lower value for the learner than the reference on the sampled states. $\mathcal{E}_{\mathcal{D}_M}^w$ is the Bellman surrogate to encourage the Q-functions to be consistent with the model-generated data $\mathcal{D}_M$. We use the double Q residual algorithm loss similar to Cheng et al. (2022), which is defined as a convex combination of the temporal difference losses with respect to the critic and the delayed target networks, $\mathcal{E}_{\mathcal{D}}^{td}(f, f', M, \pi) := \mathbb{E}_{\mathcal{D}}\left[(f(s, a) - r - \gamma f'(s', \pi))^2\right]$. $\mathcal{E}_{\mathcal{D}}(M)$ is the model-fitting loss that ensures the model is data-consistent. $\beta$ and $\lambda$ control the effect of the pessimistic loss, by constraining Q-functions and models the adversary can choose. Once the adversary is updated, we update the policy (Line 6) to maximize the pessimistic loss as defined in Eq. (5). Similar to Cheng et al. (2022), we choose one Q-function and a slower learning rate for the policy updates ($\eta_{\mathsf{fast}} \gg \eta_{\mathsf{slow}}$).

We remark that $\mathcal{E}_{\mathcal{D}_M}^w$ not only affects $f_1, f_2$, but also $M$, i.e., it forces the model to generate transitions where the Q-function is Bellman consistent. This allows the pessimistic loss to indirectly affect the model learning, thus making the model adversarial. Consider the special case where $\lambda = 0$ in the loss of Line 4. The model here is no longer forced to be data consistent, and the adversary can now freely update the model via $\mathcal{E}_{\mathcal{D}_M}^w$ such that the Q-function is always Bellman consistent. As a consequence, the algorithm becomes equivalent to IL on the model-generated states. We empirically study this behavior in our experiments (Section 5).

Lines 7 and 8 describe our model-based rollout procedure. We incrementally rollout both $\pi$ and $\pi_{\mathsf{ref}}$ from states in $\mathcal{D}_{\mathsf{real}}^{\mathsf{mini}}$ for a horizon $H$, and add the generated transitions to $\mathcal{D}_{\mathsf{model}}$. The aim of this strategy is to generate a distribution with large coverage for training the adversary and policy, and we discuss this in detail in the next section.

Finally, it is important to note the fact that neither the pessimistic nor the Bellman surrogate losses uses the real transitions; hence our algorithm is completely model-based from a statistical point of view, that the value function $f$ is solely an intermediate variable that helps in-model optimization and not directly fit from data.

## 5 Experiments

We test the efficacy of ARMOR on two major fronts: (1) performance comparison to existing offline RL algorithms, and (2) robust policy improvement over a reference policy that is not covered by the dataset, a novel setting that is not applicable to existing works[4]. We use the D4RL (Fu et al., 2020) continuous control benchmarks datasets for all our experiments and the code will be made public.

**Experimental Setup:** We parameterize $\pi, f_1, f_2$ and $M$ using feedforward neural networks, and set $\eta_{\mathsf{fast}} = 5e-4$, $\eta_{\mathsf{slow}} = 5e-7$, $w = 0.5$ similar to Cheng et al. (2022). In all our experiments, we vary only the $\beta$ and $\lambda$ parameters which control the amount of pessimism; others are fixed. Importantly, we set the rollout horizon to be the max episode horizon defined in the environment.

---

[4]In Appendix F we empirically show how imitation learning can be obtained as a special case of ARMOR

| Dataset | ARMOR | MoREL | MOPO | RAMBO | COMBO | ATAC | CQL | IQL | BC |
|---|---|---|---|---|---|---|---|---|---|
| hopper-med | **101.4** | **95.4** | 28.0 | **92.8** | 97.2 | 85.6 | 86.6 | 66.3 | 29.0 |
| walker2d-med | **90.7** | 77.8 | 17.8 | **86.9** | 81.9 | **89.6** | 74.5 | 78.3 | 6.6 |
| halfcheetah-med | 54.2 | 42.1 | 42.3 | **77.6** | 54.2 | 53.3 | 44.4 | 47.4 | 36.1 |
| hopper-med-replay | **97.1** | **93.6** | 67.5 | **96.6** | 89.5 | **102.5** | 48.6 | **94.7** | 11.8 |
| walker2d-med-replay | **85.6** | 49.8 | 39.0 | **85.0** | 56.0 | **92.5** | 32.6 | 73.9 | 11.3 |
| halfcheetah-med-replay | 50.5 | 40.2 | 53.1 | **68.9** | 55.1 | 48.0 | 46.2 | 44.2 | 38.4 |
| hopper-med-exp | **103.4** | **108.7** | 23.7 | 83.3 | **111.1** | **111.9** | **111.0** | 91.5 | **111.9** |
| walker2d-med-exp | **112.2** | 95.6 | 44.6 | 68.3 | 103.3 | **114.2** | 98.7 | **109.6** | 6.4 |
| halfcheetah-med-exp | 93.5 | 53.3 | 63.3 | **93.7** | 90.0 | **94.8** | 62.4 | 86.7 | 35.8 |
| pen-human | **72.8** | - | - | - | - | 53.1 | 37.5 | **71.5** | 34.4 |
| hammer-human | 1.9 | - | - | - | - | 1.5 | **4.4** | 1.4 | 1.5 |
| door-human | 6.3 | - | - | - | - | 2.5 | **9.9** | 4.3 | 0.5 |
| relocate-human | **0.4** | - | - | - | - | 0.1 | 0.2 | 0.1 | 0.0 |
| pen-cloned | **51.4** | - | - | - | - | 43.7 | 39.2 | 37.3 | **56.9** |
| hammer-cloned | 0.7 | - | - | - | - | 1.1 | **2.1** | **2.1** | 0.8 |
| door-cloned | -0.1 | - | - | - | - | **3.7** | 0.4 | 1.6 | -0.1 |
| relocate-cloned | -0.0 | - | - | - | - | **0.2** | -0.1 | -0.2 | -0.1 |
| pen-exp | 112.2 | - | - | - | - | **136.2** | 107.0 | - | 85.1 |
| hammer-exp | **118.8** | - | - | - | - | 126.9 | 86.7 | - | 125.6 |
| door-exp | **98.7** | - | - | - | - | **99.3** | 101.5 | - | 34.9 |
| relocate-exp | **96.0** | - | - | - | - | **99.4** | 95.0 | - | 101.3 |

Table 1: Performance comparison of ARMOR against baselines on the D4RL datasets. The values for ARMOR denote last iteration performance averaged over 4 random seeds, and baseline values were taken from their respective papers. The values denote normalized returns based on random and expert policy returns similar to Fu et al. (2020). Boldface denotes performance within 10% of the best performing algorithm. We report results with standard deviations in Appendix F.

The dynamics model is pre-trained for 100k steps using model-fitting loss on the offline dataset. ARMOR is then trained for 1M steps on each dataset. Refer to Appendix F for more details.

## 5.1 Comparison with Offline RL Baselines

By setting the reference policy to the behavior-cloned policy on the offline dataset, we can use ARMOR as a standard offline RL algorithm. Table 1 shows a comparison of the performance of ARMOR against SoTA model-free and model-based offline RL baselines. In the former category, we consider ATAC (Cheng et al., 2022), CQL (Kumar et al., 2020) and IQL (Kostrikov et al., 2021), and for the latter we consider MoREL (Kidambi et al., 2020), MOPO (Yu et al., 2020), and RAMBO (Rigter et al., 2022). We also compare against COMBO (Yu et al., 2021) which is a hybrid model-free and model-based algorithm. In these experiments, we initially warm start the optimization for 100k steps, by training the policy and Q-function using behavior cloning and temporal difference learning respectively on the offline dataset to ensure the learner policy is initialized to be the same as the reference. Overall, we observe that ARMOR consistently outperforms or is competitive with the best baseline algorithm on most datasets. Specifically, compared to other purely model-based baselines (MoREL, MOPO and RAMBO), there is a marked increase in performance in the *walker2d-med, hopper-med-exp* and *walker2d-med-exp* datasets. We would like to highlight two crucial elements about ARMOR, in contrast to other model-based baselines - (1) ARMOR achieves SoTA performance using only a *single* neural network to model the MDP, as opposed to complex network ensembles employed in previous model-based offline RL methods (Kidambi et al., 2020; Yu et al., 2021, 2020; Rigter et al., 2022), and (2) to the best of our knowledge, ARMOR is the only purely model-based offline RL algorithm that has shown performance comparable with model-free algorithms on the high-dimensional Adroit environments. The lower performance compared to RAMBO on *halfcheetah-med* and *halfcheetah-med-replay* may be attributed to that the much larger computational budget used by RAMBO is required for convergence on these datasets.

## 5.2 Robust Policy Improvement

Next, we test whether the practical version of ARMOR demonstrates RPI of the theoretical version. We consider a set of 14 datasets comprised of the *medium* and *medium-replay* versions of D4RL locomotion tasks, as well as the *human* and *cloned* versions of the Adroit tasks, with the reference policy set to be the stochastic behavior cloned policy on the expert dataset. We chose these combi-

---

[5]The variation in performance of the reference for different dataset qualities in the same environment is owing to different random seeds.

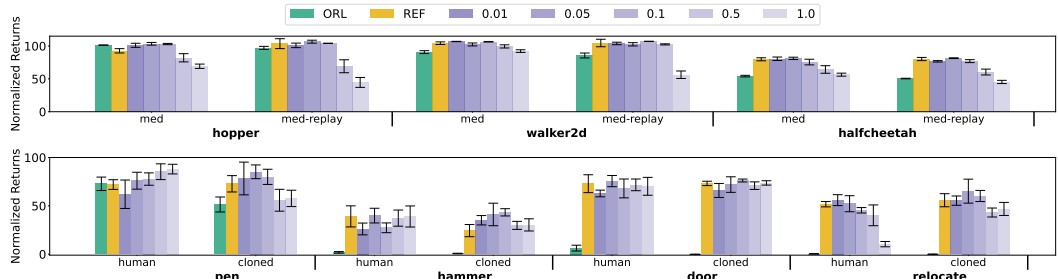

Figure 3: Verification of RPI over the reference policy for different $\beta$ (purple). ORL denotes the performance of offline RL with ARMOR ( Table 1), and REF is the performance of reference policy.[5]

nations of dataset quality and reference, to ensure that the reference policy takes out-of-distribution actions with respect to the data. Unlike Sec. 5.1 here the reference policy is a black-box given as a part of the problem definition. This opens the question of how the learner should be initialized, since we can not trivially initialize the learner to be the reference as in the previous experiments.[6] In a similar spirit to Sec. 5.1, one might consider initializing the learner close to the reference by behavior cloning the reference policy on the provided dataset during warmstart, i.e, by replacing the dataset actions with reference actions. However, when the reference chooses out of support actions, this procedure will not provide a good global approximation of the reference policy, which can make the optimization problem harder. Instead, we propose to learn a residual policy where the learned policy outputs an additive correction to the reference (Silver et al., 2018). This is an appropriate choice since ARMOR does not make any restrictive assumptions about the structure of the policy class. Figure 3 shows the normalized return achieved by ARMOR for different $\beta$, with fixed values for remaining hyperparameters. We observe that ARMOR is able to achieve performance comparable or better than the reference policy for a range of $\beta$ values uniformly across all datasets, thus verifying the RPI property in practice. Specifically, there is significant improvement via RPI in the *hammer*, *door* and *relocate* domains, where running ARMOR as a pure offline RL algorithm(Section 5.1) does not show any progress [7]. Overall, we note the following metrics:

- In 14/14 datasets, ARMOR shows RPI (i.e., ARMOR policy is no worse than the reference when measured by overlap of confidence intervals). Further, considering the difference between ORL and REF as a rough indication of whether the reference is within data support, we note that in 12/14 cases REF is strictly better than ORL, and in all those cases ARMOR demonstrates RPI.

- In 5/14 datasets, the ARMOR policy is strictly better than the reference. (Criterion: the lower confidence of ARMOR performance is better than upper confidence of REF). It is important to note that this metric is highly dependent on the quality of the reference policy. Since the reference is near-expert, it can be hard for some environments to improve significantly over it.

## 6   Discussion

The RPI of ARMOR is highly valuable as it allows easy tuning of the pessimism hyperparameter without performance degradation. We believe that leveraging this property can pave the way for real-world deployment of offline RL. Thus, we next present a discussion of RPI.[8]

*When does RPI actually improve over the reference policy?*

Given ARMOR's ability to improve over an arbitrary policy, the following question naturally arises: Can ARMOR nontrivially improve the output policy of other offline algorithms, including itself? If this were true, can we repeatedly run ARMOR to improve over itself and obtain the *best* policy any algorithm can learn offline? Unfortunately, the answer is negative. Not only can ARMOR not

---

[6]In Appendix F.5 we provide further experiments for different choices of reference policies.

[7]We provide comparisons when using a behavior cloning initialization for the learner in Appendix F.

[8]Due to space limit, we defer the complete discussion to Appendix D and only provide salient points here.

improve over itself, but it also cannot improve over a variety of algorithms (e.g., absolute pessimism or minimax regret). In fact, the optimal policy of an *arbitrary* model in the version space $\mathcal{M}_\alpha$ is provably unimprovable ( Corollary 10; Appendix D). With a deep dive into when RPI gives nontrivial improvement (Appendix D), we found some interesting observations, which we highlight here.

**Return maximization and regret minimization are *different* in offline RL** These objectives generally produce different policies, even though they are equivalent in online RL. Their equivalence in online RL relies on the fact that online exploration can eventually resolve any uncertainty. In offline RL with an arbitrary data distribution, there will generally be model uncertainty that cannot be resolved, and the worst-case reasoning over such model uncertainty (i.e., $\mathcal{M}_\alpha$) leads to definitions that are no longer equivalent. Moreover, it is impossible to compare return maximization and regret minimization and make a claim about which is better. *They are not simply an algorithm design choice, but are definitions of the learning goals and guarantees themselves*—and are thus incomparable: if we care about obtaining a guarantee for the worst-case return, the return maximization is optimal by definition; if we are more interested in a guarantee for the worst-case regret, then regret minimization is optimal. We also note that analyzing algorithms under a metric that is different from the one they are designed for can lead to unusual conclusions, e.g., Xiao et al. (2021) show that optimistic/neutral/pessimistic algorithms are equally minimax-optimal in terms of their regret guarantees in offline multi-armed bandits. However, the algorithms they consider are optimistic/pessimistic with respect to the *return* (as commonly considered in the offline RL literature) not the *regret* which is the performance metric they are interested in analyzing.

$\pi_{\text{ref}}$ **is more than a hyperparameter—it defines the performance metric and learning goal** Corollary 10 in Appendix D shows that ARMOR has many different fixed points: when $\pi_{\text{ref}}$ is chosen from these fixed points, the solution to Eq. (1) is also $\pi_{\text{ref}}$. Furthermore, some of them may seem quite unreasonable for offline learning (e.g., the greedy policy to an arbitrary model in $\mathcal{M}_\alpha$ or even the optimistic policy). This is not a defect of the algorithm. Rather, because of the unresolvable uncertainty in the offline setting, there are many different performance metrics/learning goals that are generally incompatible/incomparable, and the agent designer must make a conscious choice among them and convey the intention to the algorithm. In ARMOR, such a choice is explicitly conveyed by $\pi_{\text{ref}}$, which makes ARMOR subsume return maximization and regret minimization as special cases.

## 7 Conclusion

We have presented a model-based offline RL framework, ARMOR, that can improve over arbitrary reference policies regardless of data coverage, by using the concept of relative pessimism. ARMOR provides strong theoretical guarantees with general function approximators, and exhibits robust policy improvement over the reference policy for a wide range of hyper-parameters. We have also presented a scalable deep learning instantiation of the theoretical algorithm. Empirically, we demonstrate that ARMOR indeed enjoys the RPI property, and has competitive performance with several SoTA model-free and model-based offline RL algorithms, while employing a simpler model architecture (a single MDP model) than other model-based baselines that rely on ensembles. This also opens the opportunity to leverage high-capacity world models (Hafner et al., 2023) with offline RL in the future. However, there are also some **limitations**. While RPI holds for the pessimism parameter, the others still need to be tuned. In practice, the non-convexity of the optimization can also make solving the two-player game challenging. For instance, if the adversary is not strong enough (i.e., far from solving the inner minimization), RPI would break. Further, runtime of ARMOR is slightly slower than model-free algorithms owing to extra computations for model rollouts.

## Acknowledgments and Disclosure of Funding

Nan Jiang acknowledges funding support from NSF IIS-2112471 and NSF CAREER IIS-214178.

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

# A Proofs for Section 3

## A.1 Technical Tools

**Lemma 4** (Simulation lemma). *Consider any two MDP model $M$ and $M'$, and any $\pi : \mathcal{S} \to \Delta(\mathcal{A})$, we have*

$$|J_M(\pi) - J_{M'}(\pi)| \le \frac{V_{\max}}{1 - \gamma} \mathbb{E}_{d^\pi} \left[ D_{\text{TV}} \left( P_M(\cdot \mid s, a), P_{M'}(\cdot \mid s, a) \right) \right] + \frac{1}{1 - \gamma} \mathbb{E}_{d^\pi} \left[ |R_M(s, a) - R_{M'}(s, a)| \right].$$

Lemma 4 is the standard simulation lemma in model-based reinforcement learning literature, and its proof can be found in, e.g., Uehara and Sun (2021, Lemma 7).

## A.2 MLE Guarantees

We use $\ell_{\mathcal{D}}(M)$ to denote the likelihood of model $M = (P, R)$ with offline data $\mathcal{D}$, where

$$\ell_{\mathcal{D}}(M) = \prod_{(s,a,r,s') \in \mathcal{D}} P_M(s' \mid s, a). \tag{8}$$

For the analysis around maximum likelihood estimation, we largely follow the proving idea of Agarwal et al. (2020); Liu et al. (2022), which is inspired by Zhang (2006).

The next lemma shows that the ground truth model $M^\star$ has a comparable log-likelihood compared with MLE solution.

**Lemma 5.** *Let $M^\star$ be the ground truth model. Then, with probability at least $1 - \delta$, we have*

$$\max_{M \in \mathcal{M}} \log \ell_{\mathcal{D}}(M) - \log \ell_{\mathcal{D}}(M^\star) \le \log(|\mathcal{M}|/\delta).$$

***Proof of Lemma 5.*** The proof of this lemma is obtained by a standard argument of MLE (see, e.g., van de Geer, 2000). For any $M \in \mathcal{M}$,

$$
\begin{aligned}
\mathbb{E}\left[ \exp\left( \log \ell_{\mathcal{D}}(M) - \log \ell_{\mathcal{D}}(M^\star) \right) \right] &= \mathbb{E}\left[ \frac{\ell_{\mathcal{D}}(M)}{\ell_{\mathcal{D}}(M^\star)} \right] \\
&= \mathbb{E}\left[ \frac{\prod_{(s,a,r,s') \in \mathcal{D}} P_M(s' \mid s, a)}{\prod_{(s,a,r,s') \in \mathcal{D}} P_{M^\star}(s' \mid s, a)} \right] \\
&= \mathbb{E}\left[ \prod_{(s,a,r,s') \in \mathcal{D}} \frac{P_M(s' \mid s, a)}{P_{M^\star}(s' \mid s, a)} \right] \\
&= \mathbb{E}\left[ \prod_{(s,a) \in \mathcal{D}} \mathbb{E}\left[ \frac{P_M(s' \mid s, a)}{P_{M^\star}(s' \mid s, a)} \,\middle|\, s, a \right] \right] \\
&= \mathbb{E}\left[ \prod_{(s,a) \in \mathcal{D}} \sum_{s'} P_M(s' \mid s, a) \right] \\
&= 1. \tag{9}
\end{aligned}
$$

Then by Markov's inequality, we obtain

$$
\begin{aligned}
&\mathbb{P}\left[ (\log \ell_{\mathcal{D}}(M) - \log \ell_{\mathcal{D}}(M^\star)) > \log(1/\delta) \right] \\
&\le \underbrace{\mathbb{E}\left[ \exp\left( \log \ell_{\mathcal{D}}(M) - \log \ell_{\mathcal{D}}(M^\star) \right) \right]}_{=1 \text{ by Eq. (9)}} \cdot \exp\left[ -\log(1/\delta) \right] = \delta.
\end{aligned}
$$

Therefore, taking a union bound over $\mathcal{M}$, we obtain

$$\mathbb{P}\left[ (\log \ell_{\mathcal{D}}(M) - \log \ell_{\mathcal{D}}(M^\star)) > \log(|\mathcal{M}|/\delta) \right] \le \delta.$$

This completes the proof. $\qquad\qquad\square$

The following lemma shows that, the on-support error of any model $M \in \mathcal{M}$ can be captured via its log-likelihood (by comparing with the MLE solution).

**Lemma 6.** *For any model $M$, we have with probability at least $1 - \delta$,*

$$\mathbb{E}_\mu \left[ D_{\mathrm{TV}} \left( P_M(\cdot \mid s, a), P_{M^\star}(\cdot \mid s, a) \right)^2 \right] \leq \mathcal{O} \left( \frac{\log \ell_{\mathcal{D}}(M^\star) - \log \ell_{\mathcal{D}}(M) + \log(|\mathcal{M}|/\delta)}{n} \right),$$

*where $\ell_{\mathcal{D}}(\cdot)$ is defined in Eq. (8).*

***Proof of Lemma 6.*** By Agarwal et al. (2020, Lemma 25), we have

$$\mathbb{E}_\mu \left[ D_{\mathrm{TV}} \left( P_M(\cdot \mid s, a), P_{M^\star}(\cdot \mid s, a) \right)^2 \right] \leq -2 \log \mathbb{E}_{\mu \times P_{M^\star}} \left[ \exp \left( -\frac{1}{2} \log \left( \frac{P_{M^\star}(s' \mid s, a)}{P_M(s' \mid s, a)} \right) \right) \right],$$
(10)

where $\mu \times P_{M^\star}$ denote the ground truth offline joint distribution of $(s, a, s')$.

Let $\widetilde{\mathcal{D}} = \{(\widetilde{s}_i, \widetilde{a}_i, \widetilde{r}_i, \widetilde{s}_i')\}_{i=1}^n \sim \mu$ be another offline dataset that is independent to $\mathcal{D}$. Then,

$$-n \cdot \log \mathbb{E}_{\mu \times P_{M^\star}} \left[ \exp \left( -\frac{1}{2} \log \left( \frac{P_{M^\star}(s' \mid s, a)}{P_M(s' \mid s, a)} \right) \right) \right]$$

$$= -\sum_{i=1}^n \log \mathbb{E}_{(\widetilde{s}_i, \widetilde{a}_i, \widetilde{s}_i') \sim \mu} \left[ \exp \left( -\frac{1}{2} \log \left( \frac{P_{M^\star}(\widetilde{s}_i' \mid \widetilde{s}_i, \widetilde{a}_i)}{P_M(\widetilde{s}_i' \mid \widetilde{s}_i, \widetilde{a}_i)} \right) \right) \right]$$

$$= -\log \mathbb{E}_{\widetilde{\mathcal{D}} \sim \mu} \left[ \exp \left( \sum_{i=1}^n -\frac{1}{2} \log \left( \frac{P_{M^\star}(\widetilde{s}_i' \mid \widetilde{s}_i, \widetilde{a}_i)}{P_M(\widetilde{s}_i' \mid \widetilde{s}_i, \widetilde{a}_i)} \right) \right) \,\middle|\, \mathcal{D} \right]$$

$$= -\log \mathbb{E}_{\widetilde{\mathcal{D}} \sim \mu} \left[ \exp \left( \sum_{(s,a,s') \in \widetilde{\mathcal{D}}} -\frac{1}{2} \log \left( \frac{P_{M^\star}(s' \mid s, a)}{P_M(s' \mid s, a)} \right) \right) \,\middle|\, \mathcal{D} \right].$$
(11)

We use $\ell_M(s, a, s')$ as the shorthand of $-\frac{1}{2} \log \left( \frac{P_{M^\star}(s' \mid s, a)}{P_M(s' \mid s, a)} \right)$, for any $(s, a, s') \in \mathcal{S} \times \mathcal{A} \times \mathcal{S}$. By Agarwal et al. (2020, Lemma 24) (see also Liu et al., 2022, Lemma 15), we know

$$\mathbb{E}_{\mathcal{D} \sim \mu} \left[ \exp \left( \sum_{(s,a,s') \in \mathcal{D}} \ell_M(s, a, s') - \log \mathbb{E}_{\widetilde{\mathcal{D}} \sim \mu} \left[ \exp \left( \sum_{(s,a,s') \in \widetilde{\mathcal{D}}} \ell_M(s, a, s') \right) \,\middle|\, \mathcal{D} \right] - \log |\mathcal{M}| \right) \right] \leq 1.$$

Thus, we can use Chernoff method as well as a union bound on the equation above to obtain the following exponential tail bound: with probability at least $1 - \delta$, we have for all $(P, R) = M \in \mathcal{M}$,

$$-\log \mathbb{E}_{\widetilde{\mathcal{D}} \sim \mu} \left[ \exp \left( \sum_{(s,a,s') \in \widetilde{\mathcal{D}}} \ell_M(s, a, s') \right) \,\middle|\, \mathcal{D} \right] \leq -\sum_{(s,a,s') \in \mathcal{D}} \ell_M(s, a, s') + 2 \log(|\mathcal{M}|/\delta).$$
(12)

Plugging back the definition of $\ell_M$ and combining Eqs. (10) to (12), we obtain

$$n \cdot \mathbb{E}_\mu \left[ D_{\mathrm{TV}} \left( P(\cdot \mid s, a), P_{M^\star}(\cdot \mid s, a) \right)^2 \right] \leq \frac{1}{2} \sum_{(s,a,s') \in \mathcal{D}} \log \left( \frac{P_{M^\star}(s' \mid s, a)}{P(s' \mid s, a)} \right) + 2 \log(|\mathcal{M}|/\delta).$$

Therefore, we obtain

$$n \cdot \mathbb{E}_\mu \left[ D_{\mathrm{TV}} \left( P(\cdot \mid s, a), P_{M^\star}(\cdot \mid s, a) \right)^2 \right]$$

$$\lesssim \sum_{(s,a,s') \in \mathcal{D}} \log \left( \frac{P_{M^\star}(s' \mid s, a)}{P(s' \mid s, a)} \right) + \log(|\mathcal{M}|/\delta)$$

$$= \log \ell_{\mathcal{D}}(M^\star) - \log \ell_{\mathcal{D}}(M) + \log(|\mathcal{M}|/\delta). \qquad (\ell_{\mathcal{D}}(\cdot) \text{ is defined in Eq. (8)})$$

This completes the proof. $\qquad\qquad\square$

### A.3 Guarantees about Model Fitting Loss

**Lemma 7.** *Let $M^\star$ be the ground truth model. Then, with probability at least $1 - \delta$, we have*

$$\mathcal{E}_{\mathcal{D}}(M^\star) - \min_{M \in \mathcal{M}} \mathcal{E}_{\mathcal{D}}(M) \leq \mathcal{O}\left(\log(|\mathcal{M}|/\delta)\right),$$

*where $\mathcal{E}_{\mathcal{D}}$ is defined in Eq. (3).*

**Proof of Lemma 7.** By defition, we know

$$\mathcal{E}_{\mathcal{D}}(M) = -\log \ell_{\mathcal{D}}(M) + {(R_M(s,a)-r)^2}/{V_{\max}^2}$$

By Lemma 5, we know

$$\max_{M \in \mathcal{M}} \log \ell_{\mathcal{D}}(M) - \log \ell_{\mathcal{D}}(M^\star) \leq \log(|\mathcal{M}|/\delta). \tag{13}$$

In addition, by Xie et al. (2021a, Theorem A.1) (with setting $\gamma = 0$), we know w.p. $1 - \delta$,

$$\sum_{(s,a,r,s') \in \mathcal{D}} (R^\star(s,a) - r)^2 - \min_{M \in \mathcal{M}} \sum_{(s,a,r,s') \in \mathcal{D}} (R_M(s,a) - r)^2 \lesssim \log(|\mathcal{M}|/\delta). \tag{14}$$

Combining Eqs. (13) and (14) and using the fact of $V_{\max} \geq 1$, we have w.p. $1 - \delta$,

$$\mathcal{E}_{\mathcal{D}}(M^\star) - \min_{M \in \mathcal{M}} \mathcal{E}_{\mathcal{D}}(M)$$

$$\leq \max_{M \in \mathcal{M}} \log \ell_{\mathcal{D}}(M) - \min_{M \in \mathcal{M}} \sum_{(s,a,r,s') \in \mathcal{D}} {(R_M(s,a)-r)^2}/{V_{\max}^2} + \mathcal{E}_{\mathcal{D}}(M^\star)$$

$$\lesssim \log(|\mathcal{M}|/\delta).$$

This completes the proof. $\qquad\square$

**Lemma 8.** *For any $M \in \mathcal{M}$, we have with probability at least $1 - \delta$,*

$$\mathbb{E}_\mu\left[D_{\mathrm{TV}}\left(P_M(\cdot \mid s,a), P_{M^\star}(\cdot \mid s,a)\right)^2 + {(R_M(s,a)-R^\star(s,a))^2}/{V_{\max}^2}\right]$$

$$\leq \mathcal{O}\left(\frac{\mathcal{E}_{\mathcal{D}}(M) - \mathcal{E}_{\mathcal{D}}(M^\star) + \log(|\mathcal{M}|/\delta)}{n}\right),$$

*where $\mathcal{E}_{\mathcal{D}}$ is defined in Eq. (3).*

**Proof of Lemma 8.** By Lemma 6, we have w.p. $1 - \delta$,

$$n \cdot \mathbb{E}_\mu\left[D_{\mathrm{TV}}\left(P_M(\cdot \mid s,a), P_{M^\star}(\cdot \mid s,a)\right)^2\right] \lesssim \log \ell_{\mathcal{D}}(M^\star) - \log \ell_{\mathcal{D}}(M) + \log(|\mathcal{M}|/\delta). \tag{15}$$

Also, we have

$$n \cdot \mathbb{E}_\mu\left[(R_M(s,a) - R^\star(s,a))^2\right] \tag{16}$$

$$= n \cdot \mathbb{E}_\mu\left[(R_M(s,a) - r)^2\right] - n \cdot \mathbb{E}_\mu\left[(R^\star(s,a) - r)^2\right]$$

$$\text{(see, e.g., Xie et al., 2021a, Eq. (A.10) with } \gamma = 0)$$

$$\lesssim \sum_{(s,a,r,s') \in \mathcal{D}} (R_M(s,a) - r)^2 - \sum_{(s,a,r,s') \in \mathcal{D}} (R^\star(s,a) - r)^2 + \log(|\mathcal{M}|/\delta),$$

where the last inequality is a direct implication of Xie et al. (2021a, Lemma A.4). Combining Eqs. (15) and (16) and using the fact of $V_{\max} \geq 1$, we obtain

$$n \cdot \mathbb{E}_\mu\left[D_{\mathrm{TV}}\left(P_M(\cdot \mid s,a), P_{M^\star}(\cdot \mid s,a)\right)^2 + {(R_M(s,a)-R^\star(s,a))^2}/{V_{\max}^2}\right]$$

$$\lesssim \log \ell_{\mathcal{D}}(M^\star) - \sum_{(s,a,r,s') \in \mathcal{D}} {(R^\star(s,a)-r)^2}/{V_{\max}^2} - \log \ell_{\mathcal{D}}(M) + \sum_{(s,a,r,s') \in \mathcal{D}} {(R_M(s,a)-r)^2}/{V_{\max}^2} + \log(|\mathcal{M}|/\delta)$$

$$= \mathcal{E}_{\mathcal{D}}(M) - \mathcal{E}_{\mathcal{D}}(M^\star) + \log(|\mathcal{M}|/\delta).$$

This completes the proof. $\qquad\square$

## A.4 Proof of Main Theorems

***Proof of Theorem 2.*** By the optimality of $\widehat{\pi}$ (from Eq. (1)), we have

$$
\begin{aligned}
J(\pi^\dagger) - J(\widehat{\pi}) &= J(\pi^\dagger) - J(\pi_{\mathsf{ref}}) - [J(\widehat{\pi}) - J(\pi_{\mathsf{ref}})] \\
&\leq J(\pi^\dagger) - J(\pi_{\mathsf{ref}}) - \min_{M \in \mathcal{M}_\alpha} [J_M(\widehat{\pi}) - J_M(\pi_{\mathsf{ref}})] \qquad (\star) \\
&\leq J(\pi^\dagger) - J(\pi_{\mathsf{ref}}) - \min_{M \in \mathcal{M}_\alpha} [J_M(\pi^\dagger) - J_M(\pi_{\mathsf{ref}})], \qquad (17)
\end{aligned}
$$

where step $(\star)$ follows from Lemma 5 so that we have $M^\star \in \mathcal{M}_\alpha$, and the last step is because of $\pi^\dagger \in \Pi$. By the simulation lemma (Lemma 4), we know for any policy $\pi$ and any $M \in \mathcal{M}_\alpha$,

$$
\begin{aligned}
|J(\pi) - J_M(\pi)| &\leq \frac{V_{\max}}{1-\gamma} \mathbb{E}_{d^\pi} \left[ D_{\mathrm{TV}} \left( P_M(\cdot \mid s,a), P_{M^\star}(\cdot \mid s,a) \right) \right] + \frac{1}{1-\gamma} \mathbb{E}_{d^\pi} \left[ |R_M(s,a) - R^\star(s,a)| \right] \\
&\leq \frac{V_{\max}}{1-\gamma} \sqrt{\mathbb{E}_{d^\pi} \left[ D_{\mathrm{TV}} \left( P_M(\cdot \mid s,a), P_{M^\star}(\cdot \mid s,a) \right)^2 \right]} + \frac{V_{\max}}{1-\gamma} \sqrt{\mathbb{E}_{d^\pi} \left[ (R_M(s,a) - R^\star(s,a))^2/V_{\max}^2 \right]} \\
&\lesssim \frac{V_{\max}}{1-\gamma} \sqrt{\mathbb{E}_{d^\pi} \left[ D_{\mathrm{TV}} \left( P_M(\cdot \mid s,a), P_{M^\star}(\cdot \mid s,a) \right)^2 + (R_M(s,a) - R^\star(s,a))^2/V_{\max}^2 \right]} \\
&\qquad\qquad\qquad\qquad\qquad\qquad (a \lesssim b \text{ means } a \leq \mathcal{O}(b)) \\
&\leq \frac{V_{\max} \sqrt{\mathfrak{C}_{\mathcal{M}}(\pi)}}{1-\gamma} \sqrt{\mathbb{E}_{\mu} \left[ D_{\mathrm{TV}} \left( P_M(\cdot \mid s,a), P_{M^\star}(\cdot \mid s,a) \right)^2 + (R_M(s,a) - R^\star(s,a))^2/V_{\max}^2 \right]} \\
&\lesssim \frac{V_{\max} \sqrt{\mathfrak{C}_{\mathcal{M}}(\pi)}}{1-\gamma} \sqrt{\frac{\mathcal{E}_{\mathcal{D}}(M) - \mathcal{E}_{\mathcal{D}}(M^\star) + \log(|\mathcal{M}|/\delta)}{n}} \qquad \text{(by Lemma 8)} \\
&\lesssim \frac{V_{\max} \sqrt{\mathfrak{C}_{\mathcal{M}}(\pi)}}{1-\gamma} \sqrt{\frac{\mathcal{E}_{\mathcal{D}}(M) - \min_{M' \in \mathcal{M}} \mathcal{E}_{\mathcal{D}}(M') + \log(|\mathcal{M}|/\delta)}{n}} \qquad (\ddagger) \\
&\lesssim \frac{V_{\max} \sqrt{\mathfrak{C}_{\mathcal{M}}(\pi)}}{1-\gamma} \sqrt{\frac{\log(|\mathcal{M}|/\delta)}{n}} \qquad (18)
\end{aligned}
$$

where the step $(\ddagger)$ follows from the assumption of $M^\star \in \mathcal{M}$, and last step is because $\mathcal{E}_{\mathcal{D}}(M) - \min_{M' \in \mathcal{M}} \mathcal{E}_{\mathcal{D}}(M') \leq \alpha = \mathcal{O}(\log(|\mathcal{M}|/\delta)$ by Eq. (2).

Combining Eqs. (17) and (18), we obtain

$$
J(\pi^\dagger) - J(\widehat{\pi}) \lesssim \left[ \sqrt{\mathfrak{C}_{\mathcal{M}}(\pi^\dagger)} + \sqrt{\mathfrak{C}_{\mathcal{M}}(\pi_{\mathsf{ref}})} \right] \cdot \frac{V_{\max}}{1-\gamma} \sqrt{\frac{\log(|\mathcal{M}|/,\delta)}{n}}.
$$

This completes the proof. $\qquad\square$

Note that, over the proof above, only steps $(\star)$ and $(\ddagger)$ have used the realizability assumption of $M^\star \in \mathcal{M}$. To extend that to the misspecification case, where there only exists an $\widetilde{M}^\star \in \mathcal{M}$ such that $\widetilde{M}^\star$ is close to $M^\star$ up to some misspecification error, we just need the following straightforward accommodations:

(1) A variant of Lemma 5—to ensure that $\widetilde{M}^\star$ is included in the version space $\mathcal{M}_\alpha$. By doing so, the misspecification error should also be included in the radius of the version space.

(2) Upper bound $|J(\pi) - J_{\widetilde{M}^\star}(\pi)|$ for any $\pi$ using misspecification error. This is a standard argument, and by combining with the item above, $(\star)$ becomes $J(\widehat{\pi}) - J(\pi_{\mathsf{ref}}) \geq \min_{M \in \mathcal{M}_\alpha} [J_M(\widehat{\pi}) - J_M(\pi_{\mathsf{ref}})] - $ misspecification error.

(3) Upper bound difference in model-fitting error $\mathcal{E}_{\mathcal{D}}(\widetilde{M}^\star) - \mathcal{E}_{\mathcal{D}}(M^\star)$ using misspecification error. Then, step $(\ddagger)$ becomes, for $M \in \mathcal{M}_\alpha$,

$$
\begin{aligned}
\mathcal{E}_{\mathcal{D}}(M) - \mathcal{E}_{\mathcal{D}}(M^\star) &= \mathcal{E}_{\mathcal{D}}(M) - \mathcal{E}_{\mathcal{D}}(\widetilde{M}^\star) + \mathcal{E}_{\mathcal{D}}(\widetilde{M}^\star) - \mathcal{E}_{\mathcal{D}}(M^\star) \\
&\leq \mathcal{E}_{\mathcal{D}}(M) - \min_{M' \in \mathcal{M}} \mathcal{E}_{\mathcal{D}}(M') + \text{misspecification error} \\
&\lesssim \log(|\mathcal{M}|/\delta) + \text{misspecification error}.
\end{aligned}
$$

Due to the unboundedness of the likelihood, we conjecture that naively defining misspecification error using total variation without any accommodation on the MLE loss may be insufficient for the steps above. To resolve that, we may adopt an alternate misspecification definition, e.g., $\left|\log P_{M^\star}(s' \mid s, a) - \log P_{\widetilde{M}^\star}(s' \mid s, a)\right| \leq \varepsilon, \ \forall (s, a, s') \in \mathcal{S} \times \mathcal{A} \times \mathcal{S}$, or add extra smoothing to the MLE loss with regularization.

***Proof of Theorem 3.***

$$
\begin{aligned}
J(\pi_{\mathsf{ref}}) - J(\widehat{\pi}) = J(\pi_{\mathsf{ref}}) &- J(\pi_{\mathsf{ref}}) - [J(\widehat{\pi}) - J(\pi_{\mathsf{ref}})] \\
&\leq - \min_{M \in \mathcal{M}_\alpha} [J_M(\widehat{\pi}) - J_M(\pi_{\mathsf{ref}})] &\text{(by Lemma 5, we have } M^\star \in \mathcal{M}_\alpha) \\
&= - \max_{\pi \in \Pi} \min_{M \in \mathcal{M}_\alpha} [J_M(\pi) - J_M(\pi_{\mathsf{ref}})] &\text{(by the optimality of } \widehat{\pi} \text{ from Eq. (1))} \\
&\leq - \min_{M \in \mathcal{M}_\alpha} [J_M(\pi_{\mathsf{ref}}) - J_M(\pi_{\mathsf{ref}})] &(\pi_{\mathsf{ref}} \in \Pi) \\
&= 0.
\end{aligned}
$$

$\square$

A misspecified version of Theorem 3 can be derived similarly to what we discussed about that of Theorem 2. If the policy class is also misspecified, where there exists only $\widetilde{\pi}_{\mathsf{ref}} \in \Pi$ that is close to $\pi_{\mathsf{ref}}$ up to some misspecification error, the second last step of the proof of Theorem 3 becomes $- \min_{M \in \mathcal{M}_\alpha} [J_M(\widetilde{\pi}_{\mathsf{ref}}) - J_M(\pi_{\mathsf{ref}})] \leq$ misspecification error by simply applying the performance difference lemma on the difference between $\widetilde{\pi}_{\mathsf{ref}}$ and $\pi_{\mathsf{ref}}$.

# B Proofs for Section 6

***Proof of Lemma 9.*** We prove the result by contradiction. First notice $\min_{M \in \mathcal{M}} J_M(\pi') - J_M(\pi') = 0$. Suppose there is $\bar{\pi} \in \Pi$ such that $\min_{M \in \mathcal{M}_\alpha} J_M(\bar{\pi}) - J_M(\pi') > 0$, which implies that $J_M(\bar{\pi}) > J_M(\pi'), \forall M \in \mathcal{M}_\alpha$. Since $\mathcal{M} \subseteq \mathcal{M}_\alpha$, we have

$$
\min_{M \in \mathcal{M}} J_M(\bar{\pi}) + \psi(M) > \min_{M \in \mathcal{M}} J_M(\pi') + \psi(M) = \max_{\pi \in \Pi} \min_{M \in \mathcal{M}} J_M(\pi) + \psi(M)
$$

which is a contradiction of the maximin optimality. Thus $\max_{\pi \in \Pi} \min_{M \in \mathcal{M}_\alpha} J_M(\bar{\pi}) - J_M(\pi') = 0$, which means $\pi'$ is a solution.

For the converse statement, suppose $\pi$ is a fixed point. We can just let $\psi(M) = -J_M(\pi)$. Then this pair of $\pi$ and $\psi$ by definition of the fixed point satisfies Eq. (19). $\square$

# C Related Work

There has been an extensive line of works on reinforcement with offline/batch data, especially for the case with the data distribution is rich enough to capture the state-action distribution for any given policy (Munos, 2003; Antos et al., 2008; Munos and Szepesvári, 2008; Farahmand et al., 2010; Lange et al., 2012; Chen and Jiang, 2019; Liu et al., 2020a; Xie and Jiang, 2020, 2021). However, this assumption is not practical since the data distribution is typically restricted by factors such as the quality of available policies, safety concerns, and existing system constraints, leading to narrower coverage. As a result, recent offline RL works in both theoretical and empirical literature have focused on systematically addressing datasets with inadequate coverage.

Modern offline reinforcement learning approaches can be broadly categorized into two groups for the purpose of learning with partial coverage. The first type of approaches rely on behavior regularization, where the learned policy is encouraged to be close to the behavior policy in states where there is insufficient data (e.g., Fujimoto et al., 2018; Laroche et al., 2019; Kumar et al., 2019; Siegel et al., 2020). These algorithms ensure that the learned policy performs at least as well as the behavior policy while striving to improve it when possible, providing a form of safe policy improvement guarantees. These and other studies (Wu et al., 2019; Fujimoto and Gu, 2021; Kostrikov et al., 2021) have provided compelling empirical evidence for the benefits of these approaches.

The second category of approaches that has gained prevalence relies on the concept of *pessimism under uncertainty* to construct lower-bounds on policy performance without explicitly constraining

the policy. Recently, there have been several model-free and model-based algorithms based on this concept that have shown great empirical performance on high dimensional continuous control tasks. Model-free approaches operate by constructing lower bounds on policy performance and then optimizing the policy with respect to this lower bound (Kumar et al., 2020; Kostrikov et al., 2021). The model-based counterparts first learn a world model and the optimize a policy using model-based rollouts via off-the-shelf algorithms such as Natural Policy Gradient (Kakade, 2001) or Soft-Actor Critic (Haarnoja et al., 2018). Pessimism is introduced by either terminating model rollouts using uncertainty estimation from an ensemble of neural network models (Kidambi et al., 2020) or modifying the reward function to penalize visiting uncertain regions (Yu et al., 2020). Yu et al. (2021) propose a hybrid model-based and model-free approach that integrates model-based rollouts into a model-free algorithm to construct tighter lower bounds on policy performance. On the more theoretical side, the offline RL approaches built upon the pessimistic concept (e.g., Liu et al., 2020b; Jin et al., 2021; Rashidinejad et al., 2021; Xie et al., 2021a; Zanette et al., 2021; Uehara and Sun, 2021; Shi et al., 2022) also illustrate desired theoretical efficacy under various of setups.

Another class of approaches employs an adversarial training framework, where offline RL is posed a two player game between an adversary that chooses the worst-case hypothesis (e.g., a value function or an MDP model) from a hypothesis class, and a policy player that tried to maximize the adversarially chosen hypothesis. Xie et al. (2021a) propose the concept of Bellman-consistent pessimism to constrain the class of value functions to be Bellman consistent on the data. Cheng et al. (2022) extend this framework by introducing a relative pessimism objective which allows for robust policy improvement over the data collection policy $\mu$ for a wide range of hyper-parameters. Our approach can be interpreted as a model-based extension of Cheng et al. (2022). These approaches provide strong theoretical guarantees even with general function approximators while making minimal assumptions about the function class (realizability and Bellman completeness). Chen et al. (2022) provide an adversarial model learning method that uses an adversarial policy to generate a data-distribution where the model performs poorly and iteratively updating the model on the generated distribution. There also exist model-based approaches based on the same principle (Uehara and Sun, 2021; Rigter et al., 2022) for optimizing the absolute performance. Of these, Rigter et al. (2022) is the closest to our approach, as they also aim to find an adversarial MDP model that minimizes policy performance. They use a policy gradient approach to train the model, and demonstrate great empirical performance. However, their approach is based on absolute pessimism and does not enjoy the same RPI property as ARMOR.

# D  A Deeper Discussion of Robust Policy Improvement

## D.1  How to formally define RPI?

Improving over some reference policy has been long studied in the literature. To highlight the advantage of ARMOR, we formally give the definition of different policy improvement properties.

**Definition 2** (Robust policy improvement). *Suppose $\widehat{\pi}$ is the learned policy from an algorithm. We say the algorithm has the policy improvement (PI) guarantee if $J(\pi_{\mathsf{ref}}) - J(\widehat{\pi}) \leq {}^{o(N)}/_{N}$ is guaranteed for some reference policy $\pi_{\mathsf{ref}}$ with offline data $\mathcal{D} \sim \mu$, where $N = |\mathcal{D}|$. We use the following two criteria w.r.t. $\pi_{\mathsf{ref}}$ and $\mu$ to define different kinds PI:*

  (i) *The PI is strong if $\pi_{\mathsf{ref}}$ can be selected arbitrarily from policy class $\Pi$ regardless of the choice data-collection policy $\mu$; otherwise, PI is weak (i.e., $\pi_{\mathsf{ref}} \equiv \mu$ is required).*

  (ii) *The PI is robust if it can be achieved by a range of hyperparameters with a known subset.*

Weak policy improvement is also known as *safe policy improvement* in the literature (Fujimoto et al., 2019; Laroche et al., 2019). It requires the reference policy to be also the behavior policy that collects the offline data. In comparison, strong policy improvement imposes a stricter requirement, which requires policy improvement *regardless* of how the data were collected. This condition is motivated by the common situation where the reference policy is not the data collection policy. Finally, since we are learning policies offline, without online interactions, it is not straightforward to tune the hyperparameter directly. Therefore, it is desirable that we can design algorithms with these properties in a robust manner in terms of hyperparameter selection. Formally, Definition 2 requires the policy improvement to be achievable by a set of hyperparameters that is known before learning.

Theorem 3 indicates the robust strong policy improvement of ARMOR. On the other hand, algorithms with robust weak policy improvement are available in the literature (Fujimoto et al., 2019; Kumar et al., 2019; Wu et al., 2019; Laroche et al., 2019; Fujimoto and Gu, 2021; Cheng et al., 2022); this is usually achieved by designing the algorithm to behave like IL for a known set of hyperparameter (e.g., behavior regularization algorithms have a weight that can turn off the RL behavior and regress to IL). However, deriving guarantees of achieving the best data-covered policy of the IL-like algorithm is challenging due to its imitating nature. To our best knowledge, ATAC (Cheng et al., 2022) is the only algorithm that achieves both robust (weak) policy improvement as well as guarantees absolute performance.

## D.2   When RPI actually improves?

Given ARMOR's ability to improve over an arbitrary policy, the following questions naturally arise: *Can ARMOR nontrivially improve the output policy of other algorithms (e.g., such as those based on* absolute pessimism *(Xie et al., 2021a)), including itself?* Note that outputting $\pi_{\mathsf{ref}}$ itself always satisfies RPI, but such result is trivial. By "nontrivially" we mean a non-zero worst-case improvement. If the statement were true, we would be able to repeatedly run ARMOR to improve over itself and then obtain the *best* policy any algorithm can learn offline.

Unfortunately, the answer is negative. Not only ARMOR cannot improve over itself, but it also cannot improve over a variety of algorithms. In fact, the optimal policy of an *arbitrary* model in the version space is unimprovable (see Corollary 10)! Our discussion reveals some interesting observations (e.g., how equivalent performance metrics for online RL can behave very differently in the offline setting) and their implications (e.g., how we should choose $\pi_{\mathsf{ref}}$ for ARMOR). Despite their simplicity, we feel that many in the offline RL community are not actively aware of these facts (and the unawareness has led to some confusion), which we hope to clarify below.

**Setup**   We consider an abstract setup where the learner is given a version space $\mathcal{M}_\alpha$ that contains the true model and needs to choose a policy $\pi \in \Pi$ based on $\mathcal{M}_\alpha$. We use the same notation $\mathcal{M}_\alpha$ as before, but emphasize that it does not have to be constructed as in Eqs. (2) and (3). In fact, for the purpose of this discussion, the data distribution, sample size, data randomness, and estimation procedure for constructing $\mathcal{M}_\alpha$ are **all irrelevant**, as our focus here is how decisions should be made with a given $\mathcal{M}_\alpha$. This makes our setup very generic and the conclusions widely applicable.

To facilitate discussion, we define the *fixed point* of ARMOR's relative pessimism step:

**Definition 3.** *Consider Eq. (1) as an operator that maps an arbitrary policy $\pi_{\mathsf{ref}}$ to $\widehat{\pi}$. A fixed point of this* relative pessimism *operator is, therefore, any policy $\pi \in \Pi$ such that $\pi \in \mathrm{argmax}_{\pi' \in \Pi} \min_{M \in \mathcal{M}_\alpha} J_M(\pi') - J_M(\pi)$.*

Given the definition, relative pessimism cannot improve over a policy if it is already a fixed point. Below we show a sufficient and necessary condition for being a fixed point, and show a number of concrete examples (some of which may be surprising) that are fixed points and thus unimprovable.

**Lemma 9** (Fixed-point Lemma). *For any $\mathcal{M} \subseteq \mathcal{M}_\alpha$ and any $\psi : \mathcal{M} \to \mathbb{R}$, consider the policy*

$$\pi \in \mathop{\mathrm{argmax}}_{\pi' \in \Pi} \min_{M \in \mathcal{M}} J_M(\pi') + \psi(M) \tag{19}$$

*Then $\pi$ is a fixed point in Definition 3. Conversely, for any fixed point $\pi$ in Definition 3, there is a $\psi : \mathcal{M} \to \mathbb{R}$ such that $\pi$ is a solution to Eq. (19).*

**Corollary 10.** *The following are fixed points of relative pessimism (Definition 3):*

1. *Absolute-pessimism policy, i.e., $\psi(M) = 0$.*

2. *Relative-pessimism policy for any reference policy, i.e., $\psi(M) = -J_M(\pi_{\mathsf{ref}})$.*

3. *Regret-minimization policy, i.e., $\psi(M) = -J_M(\pi_M^\star)$, where $\pi_M^\star \in \mathrm{argmax}_{\pi \in \Pi} J_M(\pi)$.*

4. *Optimal policy of an* arbitrary *model $M \in \mathcal{M}_\alpha$, $\pi_M^\star$, i.e., $\mathcal{M} = \{M\}$. This would include the optimistic policy, that is, $\mathrm{argmax}_{\pi \in \Pi, M \in \mathcal{M}_\alpha} J_M(\pi)$*

**Return maximization and regret minimization are *different* in offline RL**   We first note that these four examples generally produce different policies, even though some of them optimize for

objectives that are traditionally viewed as equivalent in online RL (the "worst-case over $\mathcal{M}_\alpha$" part of the definition does not matter in online RL), e.g., absolute pessimism optimizes for $J_M(\pi)$, which is the same as minimizing the regret $J_M(\pi^\star_M) - J_M(\pi)$ for a fixed $M$. However, their equivalence in online RL relies on the fact that online exploration can eventually resolve any model uncertainty when needed, so we only need to consider the performance metrics w.r.t. the true model $M = M^\star$. In offline RL with an arbitrary data distribution (since we do not make any coverage assumptions), there will generally be model uncertainty that cannot be resolved, and worst-case reasoning over such model uncertainty (i.e., $\mathcal{M}_\alpha$) separates apart the definitions that are once equivalent.

Moreover, it is impossible to compare return maximization and regret minimization and make a claim about which one is better. They are not simply an algorithm design choice, but are definitions of the learning goals and the guarantees themselves—thus incomparable: if we care about obtaining a guarantee for the worst-case *return*, the return maximization is optimal by definition; if we are more interested in obtaining a guarantee for the worst-case *regret*, then again, regret minimization is trivially optimal. We also note that analyzing algorithms under a metric that is different from the one they are designed for can lead to unusual conclusions. For example, Xiao et al. (2021) show that optimistic/neutral/pessimistic algorithms[9] are equally minimax-optimal in terms of their regret guarantees in offline multi-armed bandits. However, the algorithms they consider are optimistic/pessimistic w.r.t. the return—as commonly considered in the offline RL literature—not w.r.t. the regret which is the performance metric they are interested in analyzing.

**$\pi_{\mathsf{ref}}$ is more than a hyperparameter—it defines the performance metric and learning goal** Corollary 10 shows that ARMOR (with relative pessimism) has many different fixed points, some of which may seem quite unreasonable for offline learning, such as greedy w.r.t. an arbitrary model or even optimism (#4). From the above discussion, we can see that this is not a defect of the algorithm. Rather, in the offline setting with unresolvable model uncertainty, there are many different performance metrics/learning goals that are generally incompatible/incomparable with each other, and the agent designer must make a choice among them and convey the choice to the algorithm. In ARMOR, such a choice is explicitly conveyed by the choice of $\pi_{\mathsf{ref}}$, which subsumes return maximization and regret minimization as special cases (#2 and #3 in Corollary 10)

## E  A More Comprehensive Toy Example for RPI

We illustrate with a simple toy example why ARMOR intuitively demonstrates the RPI property even when $\pi_{\mathsf{ref}}$ is not covered by the data $\mathcal{D}$. ARMOR achieves this by *1)* learning an MDP Model, and *2)* adversarially training this MDP model to minimize the relative performance difference to $\pi_{\mathsf{ref}}$ during policy optimization. Consider a one-dimensional discrete MDP with five possible states as shown in Figure 4. The dynamics is deterministic, and the agent always starts in the center cell. The agent receives a lower reward of 0.1 in the left-most state and a high reward of 1.0 upon visiting the right-most state. Say, the agent only has access to a dataset from a sub-optimal policy that always takes the left action to receive the 0.1 reward. Further, let's say we have access to a reference policy that demonstrates optimal behavior on the true MDP by always choosing the right action to visit the right-most state. However, it is unknown a priori that the reference policy is optimal. In such a case, typical offline RL methods can only recover the sub-optimal policy from the dataset as it is the best-covered policy in the data. Now, for the sake of clarity, consider the current learner policy is same as the behavior policy, i.e it always takes the left action.

ARMOR can learn to recover the expert reference policy in this example by performing rollouts with the adversarially trained MDP model. From the realizability assumption we know that the version space of models contains the true model (i.e., $M^\star \in \mathcal{M}_\alpha$). The adversary can then choose a model from this version space where the reference policy $\pi_{\mathsf{ref}}$ maximally outperforms the learner. Note, that ARMOR does not require the true reward function to be known. In this toy example, the model selected by the adversary would be the one that not only allows the expert policy to reach the right-most state, but also predicts the highest reward for doing so. Now, optimizing to maximize relative performance difference with respect to this model will ensure that the learner can recover the expert behavior, since the only way for the learner to stay competitive with the reference policy is to mimic the reference policy in the region outside data support. In other words, the reason why ARMOR has RPI to $\pi_{\mathsf{ref}}$ is that its adversarial model training procedure can augment the original offline data with

---

[9]Incidentally, optimistic/neutral policies correspond to #4 in Corollary 10.

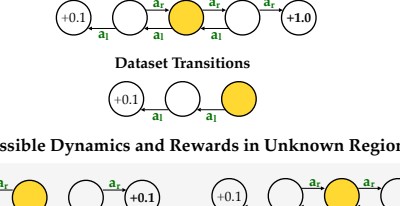

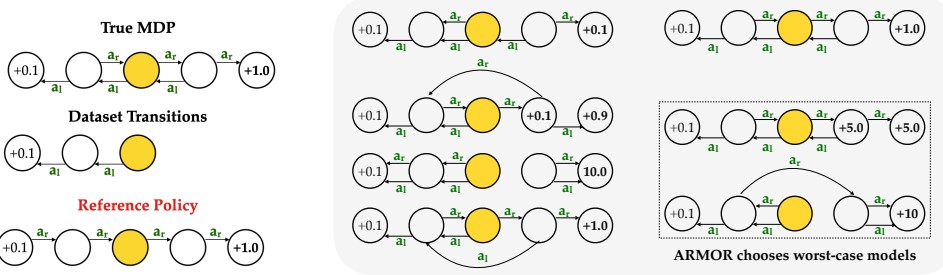

Figure 4: A toy MDP illustrating the RPI property of ARMOR. (Top) The true MDP has deterministic dynamics where taking the left ($a_l$) or right ($a_r$) actions takes the agent to corresponding states; start state is in yellow. The suboptimal behavior policy only visits only the left part of the state space, and the reference policy demonstrates optimal behavior by always choosing $a_r$. (Bottom) A subset of possible data-consistent MDP models (dynamics + rewards) in the version space. The adversary always chooses the MDP that makes the reference maximally outperform the learner. In response, the learner will learn to mimic the reference outside data support to be competitive.

new states and actions that would cover those generated by running the reference policy in the true environment, even though ARMOR does not have knowledge of $M^\star$.

# F   Further Experimental Details

## F.1   Experimental Setup and Hyper-parameters

We represent our policy $\pi$, Q-functions $f_1, f_2$ and MDP model $M$ as standard fully connected neural networks. The policy is parameterized as a Gaussian with a state-dependent covariance, and we use a tanh transform to limit the actions to the action space bound similar to Haarnoja et al. (2018). The MDP model learns to predict the next state distribution, rewards and terminal states, where the reward and next-state distributions part are parameterized as Gaussians with state-dependent covariances. The model fitting loss consists of negative log-likelihood for the next-state and reward and binary cross entropy for the terminal flags. In all our experiments we use the same model architecture and a fixed value of $\lambda$. We use Adam optimizer (Kingma and Ba, 2015) with fixed learning rates $\eta_{fast}$ and $\eta_{slow}$ similar to Cheng et al. (2022). Also similar to prior work (Kidambi et al., 2020), we let the MDP model network predict delta differences to the current state. The rollout horizon is always set to the maximum episode steps per environment. A complete list of hyper-parameters can be found in Table 3.

**Compute:** Each run of ARMOR has access to 4CPUs with 28GB RAM and a single Nvidia T4 GPU with 16GB memory. With these resources each run tasks around 6-7 hours to complete. Including all runs for 4 seeds, and ablations this amounts to approximately 2500 hours of GPU compute.

| Hyperparameter | Value |
|---|---|
| model_num_layers | 3 |
| model_hidden_size | 512 |
| model_nonlinearity | swish |
| policy_num_layers | 3 |
| policy_hidden_size | 256 |
| policy_nonlinearity | relu |
| f_num_layers | 3 |
| f_hidden_size | 256 |
| f_nonlinearity | relu |

Table 2: Model Architecture Details

| Hyperparameter | Value |
|---|---|
| critic learning rate $\eta_{\text{fast}}$ | 5e-4 |
| policy learning rate $\eta_{\text{slow}}$ | 5e-7 |
| discount factor | 0.99 |
| rollout horizon | max episode steps |
| model buffer size | $10^6$ |
| batch size | 125 |
| model batch size | 125 |
| num warmstart steps | $10^5$ |
| $\tau$ | $5e-3$ |

Table 3: List of Hyperparameters.

## F.2 Detailed Performance Comparison and RPI Ablations

In Table 4 we show the performance of ARMOR compared to model-free and model-based offline RL baselines with associate standard deviations over 8 seeds. For ablation, here we also include ARMOR$^{\dagger}$, which is running ARMOR in Algorithm 1 but without the model optimizing for the Bellman error (that is, the model is not adversarial). Although ARMOR$^{\dagger}$ does not have any theoretical guarantees (and indeed in the worst case its performance can be arbitrarily bad), we found that ARMOR$^{\dagger}$ in these experiments is performing surprisingly well. Compared with ARMOR, ARMOR$^{\dagger}$ has less stable performance when the dataset is diverse (e.g. *-med-replay* datasets) and larger learning variance. Nonetheless, ARMOR$^{\dagger}$ using a single model is already pretty competitive with other algorithms. We conjecture that this is due to that Algorithm 1 also benefits from pessimism due to adversarially trained critics. Since the model buffer would not cover all states and actions (they are continuous in these problems), the adversarially trained critic still controls the pessimism for actions not in the model buffer, as a safe guard. As a result, the algorithm can tolerate the model quality more.

| Dataset | ARMOR | ARMOR$^{\dagger}$ | ARMOR$^{re}$ | MoREL | MOPO | RAMBO | COMBO | ATAC | CQL | IQL | BC |
|---|---|---|---|---|---|---|---|---|---|---|---|
| hopper-med | **101.4 ± 0.3** | **100.4 ± 1.7** | 65.3 ± 4.8 | **95.4** | 28.0 ± 12.4 | **92.8 ± 6.0** | **97.2 ± 2.2** | 85.6 | 86.6 | 66.3 | 29.0 |
| walker2d-med | 90.7 ± 4.4 | **91.0 ± 10.4** | 79.0 ± 2.2 | 77.8 | 17.8 ± 19.3 | **86.9 ± 2.7** | **81.9 ± 2.8** | **89.6** | 74.5 | 78.3 | 6.6 |
| halfcheetah-med | 54.2 ± 2.4 | 56.3 ± 0.5 | 45.2 ± 0.2 | 42.1 | 42.3 ± 1.6 | **77.6 ± 1.5** | 54.2 ± 1.5 | 53.3 | 44.4 | 47.4 | 36.1 |
| hopper-med-replay | 97.1 ± 4.8 | 82.7 ± 23.1 | 68.4 ± 5.2 | 93.6 | 67.5 ± 24.7 | **96.6 ± 7.0** | 89.5 ± 1.8 | 102.5 | 48.6 | **94.7** | 11.8 |
| walker2d-med-replay | 85.6 ± 7.5 | 78.4 ± 1.9 | 50.3 ± 5.7 | 49.8 | 39.0 ± 9.6 | **85.0 ± 15.0** | 56.0 ± 8.6 | **92.5** | 32.6 | 73.9 | 11.3 |
| halfcheetah-med-replay | 50.5 ± 0.9 | 49.5 ± 0.9 | 36.8 ± 1.5 | 40.2 | 53.1 ± 2.0 | **68.9 ± 2.3** | 55.1 ± 1.0 | 48.0 | 46.2 | 44.2 | 38.4 |
| hopper-med-exp | 103.4 ± 5.9 | 100.1 ± 10.0 | 89.3 ± 3.2 | **108.7** | 23.7 ± 6.0 | 83.3 ± 9.1 | **111.1 ± 2.9** | **111.9** | **111.0** | 91.5 | **111.9** |
| walker2d-med-exp | **112.2 ± 1.7** | **110.5 ± 1.4** | **105.8 ± 1.4** | 95.6 | 44.6 ± 12.9 | 68.3 ± 15.0 | 103.3 ± 5.6 | **114.2** | 98.7 | **109.6** | 6.4 |
| halfcheetah-med-exp | **93.5 ± 0.5** | **93.4 ± 0.3** | 61.8 ± 3.75 | 53.3 | 63.3 ± 38.0 | **93.7 ± 10.5** | 90.0 ± 5.6 | 94.8 | 62.4 | **86.7** | 35.8 |
| pen-human | **72.8 ± 13.9** | 50.0 ± 15.6 | 62.3 ± 8.35 | - | - | - | - | 53.1 | 37.5 | **71.5** | 34.4 |
| hammer-human | 1.9 ± 1.6 | 1.1 ± 1.4 | 3.1 ± 1.9 | - | - | - | - | 1.5 | **4.4** | 1.4 | 1.5 |
| door-human | 6.3 ± 6.0 | 3.9 ± 2.4 | 5.9 ± 2.75 | - | - | - | - | 2.5 | **9.9** | 4.3 | 0.5 |
| relocate-human | **0.4 ± 0.4** | **0.4 ± 0.6** | **0.3 ± 0.25** | - | - | - | - | 0.1 | 0.2 | 0.1 | 0.0 |
| pen-cloned | 51.4 ± 15.5 | 45.2 ± 15.8 | 40.0 ± 8.25 | - | - | - | - | 43.7 | 39.2 | 37.3 | **56.9** |
| hammer-cloned | 0.7 ± 0.6 | 0.3 ± 0.0 | **2.7 ± 0.15** | - | - | - | - | 1.1 | **2.1** | **2.1** | 0.8 |
| door-cloned | -0.1 ± 0.0 | -0.1 ± 0.1 | 0.5 ± 0.4 | - | - | - | - | **3.7** | 0.4 | 1.6 | -0.1 |
| relocate-cloned | -0.0 ± 0.0 | -0.0 ± 0.0 | -0.0 ± 0.0 | - | - | - | - | **0.2** | -0.1 | -0.2 | -0.1 |
| pen-exp | 112.2 ± 6.3 | 113.0 ± 11.8 | 92.8 ± 9.25 | - | - | - | - | **136.2** | 107.0 | - | 85.1 |
| hammer-exp | 118.8 ± 5.6 | 115.3 ± 9.3 | 51.0 ± 11.05 | - | - | - | - | **126.9** | 86.7 | - | **125.6** |
| door-exp | 98.7 ± 4.1 | 97.1 ± 4.9 | 88.4 ± 3.05 | - | - | - | - | 99.3 | **101.5** | - | 34.9 |
| relocate-exp | 96.0 ± 6.8 | 90.7 ± 6.3 | 64.2 ± 7.3 | - | - | - | - | 99.4 | 95.0 | - | **101.3** |

Table 4: Performance comparison of ARMOR against baselines on the D4RL datasets. The values for ARMOR denote last iteration performance averaged over 4 random seeds along with standard deviations, and baseline values were taken from their respective papers. Boldface denotes performance within $10\%$ of the best performing algorithm.

## F.3 Effect of Residual Policy

In Figure 5, we show the effect on RPI of different schemes for initializing the learner for several D4RL datasets. Specifically, we compare using a residual policy( Section 5) versus behavior cloning the reference policy on the provided offline dataset for learner initialization. Note that this offline dataset is the suboptimal one used in offline RL and is different from the expert-level dataset used to train and produce the reference policy. We observe that using a residual policy (purple) consistently shows RPI across all datasets. However, with behavior cloning initialization (pink), there is a large variation in performance across datasets. While RPI is achieved with behavior cloning initialization on *hopper*, *walker2d* and *hammer* datasets, performance can be arbitrarily bad compared to the reference on other problems. As an ablation, we also study the effect of using a residual policy in

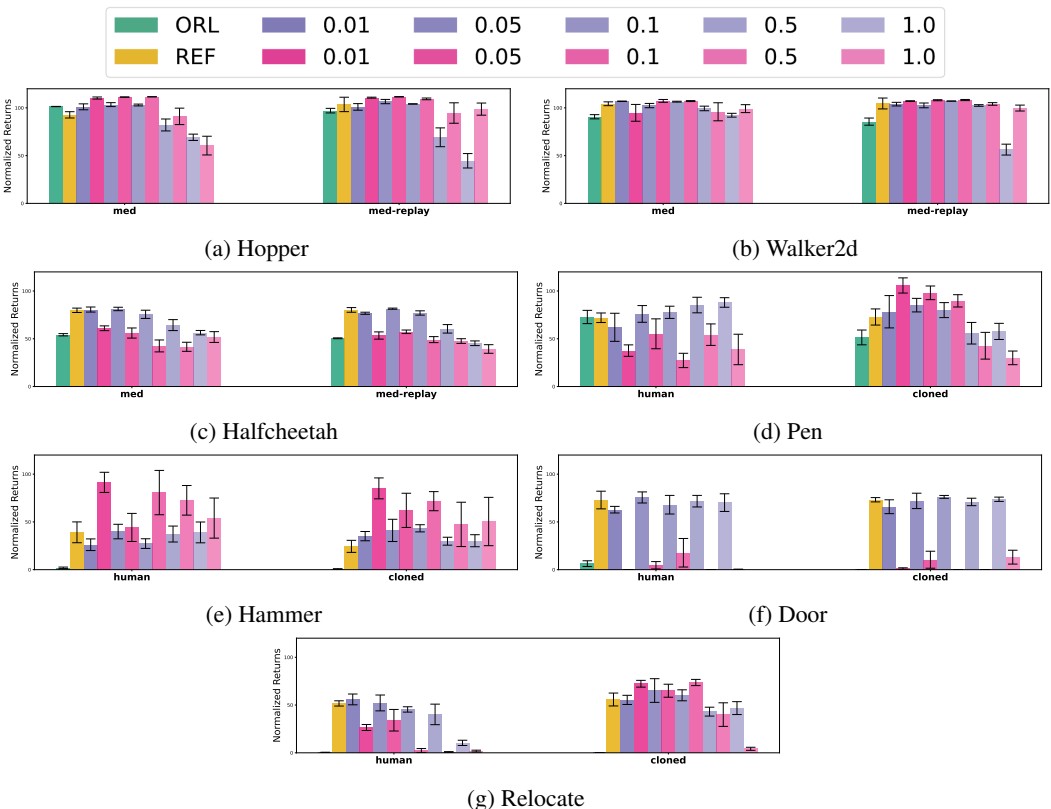

Figure 5: Comparison of different policy initializations for RPI with varying pessimism hyperparameter $\beta$. ORL denotes the performance of offline RL with ARMOR ( Table 1), and REF is the performance of reference policy. Purple represents residual policy initialization and pink is initialization using behavior cloning of the reference on the suboptimal offline RL dataset.

the offline RL case where no explicit reference is provided, and the behavior cloning policy is used as the reference similar to Section 5.1. We include the results in Table 4 as ARMOR$^{re}$, where we observe that using a residual policy overall leads to worse performance across all datasets. This lends evidence to the fact that using a residual policy is a *compromise* in instances where initializing the learner exactly to the reference policy is not possible.

## F.4    Connection to Imitation Learning

| Dataset | ARMOR-IL | BC |
|---|---|---|
| hopper-exp | 111.6 | 111.7 |
| walker2d-exp | 108.1 | 108.5 |
| halfcheetah-exp | 93.9 | 94.7 |

Table 5: ARMOR-IL on expert datasets. By setting $\lambda = 0, \beta > 0$ we recover IL.

As mentioned in Section 4.2, IL is a special case of ARMOR with $\lambda = 0$. In this setting, the Q-function can fully affect the adversarial MDP model, so the best strategy of the policy is to mimic the reference. We test this on the *expert* versions of the D4RL locomotion tasks in Table 5, and observe that ARMOR can indeed perform IL to match expert performance.

## F.5    Ablation Study: RPI for Different Reference Policies

Here we provide ablation study results for robust policy improvement under different reference policies for a wide range of $\beta$ values (pessimism hyper-parameter). For all the considered reference policies we present average normalized scroes for ARMOR and reference (REF) over multiple random seeds, and observe that ARMOR can consistently outperform the reference for a large range of $\beta$ values.

**Random Dataset Reference**  We use a reference policy obtained by running behavior cloning on the RANDOM versions of different datasets. This is equivalent to using a randomly initialized neural network as the reference.

| Dataset | 0.01 | 0.05 | 0.1 | 0.5 | 1.0 | 10.0 | 100.0 | 200.0 | 500.0 | 1000.0 | REF |
|---|---|---|---|---|---|---|---|---|---|---|---|
| hopper-med | 1.3 | 1.5 | 4.7 | 9.6 | 20.4 | 34.8 | 25.8 | 40.8 | 25.6 | 28.9 | 1.2 |
| walker2d-med | 0.0 | 0.0 | 0.2 | 1.5 | 4.0 | 17.4 | 23.6 | 12.4 | 12.1 | 20.1 | 0.0 |
| halfcheetah-med | 0.0 | 0.1 | 0.1 | 0.7 | 1.2 | -0.1 | -0.7 | 0.1 | 1.1 | -0.3 | -0.1 |
| hopper-med-replay | 1.3 | 3.0 | 8.2 | 13.3 | 39.5 | 57.6 | 48.0 | 34.4 | 51.0 | 32.9 | 1.2 |
| walker2d-med-replay | 0.0 | 0.0 | 0.1 | 4.0 | 5.9 | 13.1 | 10.8 | 14.1 | 16.5 | 16.6 | 0.0 |
| halfcheetah-med-replay | -0.2 | 0.4 | 0.3 | 0.7 | 0.9 | 7.7 | 5.8 | 8.2 | 4.9 | 6.1 | -0.2 |

**Hand-designed Reference** In this experiment we use a hand-designed reference policy called RAN-DOMBANGBANG, that selects either the minimum of maximum action in the data with 0.5 probability each.

| Dataset | 0.01 | 0.05 | 0.1 | 0.5 | 1.0 | 10.0 | 100.0 | 200.0 | 500.0 | 1000.0 | REF |
|---|---|---|---|---|---|---|---|---|---|---|---|
| hopper-med | 8.5 | 15.0 | 15.8 | 5.0 | 8.1 | 11.1 | 37.0 | 19.6 | 12.4 | 25.6 | 1.2 |
| walker2d-med | 22.7 | 36.1 | 31.6 | 42.6 | 12.0 | 55.9 | 33.6 | 37.8 | 36.6 | 49.6 | 0.1 |
| halfcheetah-med | 10.6 | 9.8 | 19.1 | 11.7 | 13.1 | 15.6 | 14.3 | 3.8 | 8.3 | 11.1 | -1.0 |
| hopper-med-replay | 33.9 | 54.8 | 62.6 | 66.7 | 55.3 | 57.0 | 67.7 | 79.5 | 70.7 | 62.0 | 1.3 |
| walker2d-med-replay | 11.6 | 35.5 | 47.5 | 40.4 | 42.7 | 47.6 | 64.8 | 49.9 | 44.7 | 33.0 | 0.1 |
| halfcheetah-med-replay | 13.1 | 10.1 | 11.3 | 9.7 | 12.4 | 17.6 | 9.0 | 14.5 | 11.1 | 9.1 | -1.3 |

