# OpenReview forum: "Adversarial Model for Offline Reinforcement Learning"
_NeurIPS.cc/2023/Conference — NeurIPS 2023 poster_

### Official Review · Reviewer_sNsB · 2023-06-18

**Soundness:** 3 good
**Presentation:** 2 fair
**Contribution:** 2 fair
**Rating:** 6
**Confidence:** 4

**Summary:**

A fundamental principle in offline RL is pessimism, which however is not free from the performance degradation with respect to the baseline reference policies, i.e., the currently running policies in the system.
In viewing this issue, this paper proposes Adversarial Model for Offline Reinforcement Learning (ARMOR), which can robustly learn policies that improve upon an arbitrary reference policy, regardless of the data quality.
Specifically, in ARMOR, the authors extend the technique of relative pessimism by adversarially training an MDP model, to achieve robust policy improvement (RPI) with arbitrary reference policies, regardless of whether they collected the data or not.
Based on the theory, a scalable implementation of ARMOR is developed, which achieves competitive performance on D4RL benchmarks, while using only a single model rather than the widely-used ensembles.




**Strengths:**

1. The proposed method is theoretically supported.
2. Practical implementation, including the selection of key hyperparameters, is discussed in details.
3. Achieving competitive performance while using only a single model worth praising, since this is important for utilizing more sophisticated dynamics model.





**Weaknesses:**

1. The assumptions, in particular the realizability of the model class $\mathcal{M}$ and $\mathcal{M_\alpha}$, and the policy class $\Pi$, maybe demanding. It is unclear if or how these assumptions holds in practice. It is also unclear if the theoretical results still hold with a misspecified model and/or policy class, e.g., $M^*  \notin \mathcal{M}$, $M^*  \notin \mathcal{M_\alpha}$ , and/or $\pi_{\mathrm{ref}} \notin \Pi$.
2. The authors draw a distinction between the reference policy and the data-collecting behavior policy. But in the experiments, the reference policy is simply set as the behavior-cloned policy on the same or a very similar offline dataset, e.g., expert v.s. medium. Therefore, this conceptual distinction seems artificial.
    * In practice/experiments, can the learned policy still improve upon the reference when the behavior policy (offline dataset) behaves significantly different from the reference policy?
    * And more importantly, why couldn't we simply learn from the data generated by the "reference policy"?
3. The authors claim that the proposed method is robust to hyperparameters within a known set, i.e., the RPI property. But there is no experiment directly showing this. I would love to see more evidence and details pertaining to, for examples,
    * How to select this set?
    * How large can this set be?
    * How many hyperparameters can this set incorporate?
4. Except the proposed method, the practical implementation (Section 4) shows many differences from the implementations of the baseline methods, such as MOPO, COMBO, CQL, IQL and so on. It is therefore unclear if the performance gain is due to the proposed method or due to these different choices of implementation details. This can be seen from the fact that the closely-related model-free method ATAC has performance similar to ARMOR and better than baselines on several datasets.

**Questions:**

1. Can existing model-based methods, such as MOPO, MoREL or COMBO, utilize the reference policy as well? For example, a quick (and naive) thought may be adding an extra behavior-cloning term towards the reference policy on model-generated states (similar to L153-155). I am curious what preventing them from doing so? Otherwise, with such an augmentation, are they still underperform ARMOR?
2. Could you elaborate more on the distinction between the reference policy and behavior policy? Are there any practical setting where the reference policy is significantly different from the policy that collect the offline dataset?
3. In Eqn. (1) and (2), why do you introduce the $\alpha$ parameter? Maybe I miss something, but I don't find $\alpha$ being used later on, except something like "for $\alpha$ large enough".



**Limitations:**

The authors adequately addressed the limitations.

---

> ### Author Rebuttal · Authors · 2023-08-10
>
>
>
> Thank you for the detailed questions. We hope that our answers below would address your concerns and clarify the importance of the contribution we are making.
>
>
> **Weakness 1**
>
> Thank you for bringing it up. We would like to emphasize that realizability is a standard assumption in the literature (see, e.g., [Uehara and Sun, 2021]). Furthermore, it is straightforward to handle the case of misspecified model and policy classes by additional additive terms. This is why we made the assumption. We will add this discussion in the camera-ready version.
>
> **Weakness 2**
>
> The distinction between the reference and the behavior policy is more than just conceptual. There are multiple practical scenarios where we can’t learn from data collected by the reference alone:
>
> 1. The data might be collected by a more exploratory policy. A priori, it's unknown whether the reference is good or safe.
> Data comes from multiple policies (e.g., a system that iterates with the environment for a couple of rounds). The behavior policy is their mixture, but ARMOR allows us to set the reference to be the best.
>
> 2. A concrete scenario is the following. Consider using offline RL to improve a recommendation system. The product team has a running policy (i.e., the baseline policy) right now and has some offline data, which is collected by previous policies and perhaps some from the current policy. Naturally, we want the policy learned by offline RL to be at least no worse than the current policy under deployment (i.e. the reference policy).  Such a scenario of learning to improve a baseline/reference policy using mixed offline data is common in offline RL applications.
>
> In Sec 5.2, we explicitly construct experiments to simulate the scenario where the reference policy is distinct from the behavior policy, and demonstrate that robust policy improvement (RPI) still holds. We found that the learned policy can indeed improve over the reference even when the reference policy (the expert) behaves differently from the behavior policy (mediocre policies); see the gap in returns. There is significant improvement especially in the halfcheetah and adroit domains. In fact, we would argue that these are the major results of the paper, and Sec 5.1, where the reference is set to the behavior cloned policy is just a demonstration of ARMOR being used as a standard offline RL algorithm when no explicit reference policy is provided.
>
>
>
> **Weakness 3**
>
> As per the theoretical results and main claims of the paper, ARMOR has robustness to the pessimism hyper-parameter $\beta$ and not all hyperparameters. Our results in Sec 5.2 validate this RPI hypothesis. The theory suggests that RPI should hold for small values of $\beta$. In practice, hence a good general strategy is to start with small $\beta$ values, and gradually increase it to find the best performing policy.
>
>
> **Weakness 4**
>
> We agree with the reviewer that this is an important question. However, in general it is hard to directly compare model-based and model-free methods over implementation details. More generally, it is difficult to study the effects of implementation details and conceptual ideas in isolation when comparing two very different deep RL algorithms. Each algorithm’s code can make different low-level design choices. Consequently, performing a strict comparison between the conceptual ideas is often impossible, unless we reimplement one algorithm based on another’s low-level details (but one might call this reimplementation yet another new deep RL algorithm?). Therefore, it’s hard to do such an isolated and direct comparison between ARMOR and other baselines like MOPO, COMBO, CQL, IQL as the reviewer mentioned.
>
>
> Nonetheless, we would like to highlight some (beneficial) effects due to the conceptual difference ARMOR introduces beyond what implementation details can provide.
>
> 1. Existing model-based methods such as MOPO and MOReL "have to" use ensembles of dynamics models for uncertainty quantification. But ARMOR does not, because it is based on adversarial training. As the reviewer mentioned, by *not* using an ensemble of models, ARMOR can be more suitable to train large models, when hosting ensembles of large models is too computationally expensive.
>
> 2. While ATAC and ARMOR share the same low-level implementation details, we show that they have different properties. ATAC only demonstrates RPI with respect to the behavior policy whereas ARMOR, due to its model-based nature has RPI with respect to arbitrary reference policies including those that are not covered by the dataset.
>
> **Question 1**
>
> This is an interesting idea but it is unclear how it can be done in a principled manner. The direction the reviewer suggested seems intuitive, but there are missing details (e.g., what the states to define the BC term in are and how to generate them) that can greatly affect the performance. Further it is unclear whether this design would have the same RPI guarantees as ARMOR in theory, as our current proof relies on the properties of adversarial training.
>
>
> **Question 2**
>
> Please refer to the response for Weakness 2.
>
> **Question 3**
>
> This $\alpha$ parameter is to account for the finite-sample errors arising from estimating the true expectation with samples. It's a standard slackness parameter in forming version space (see, e.g., $\xi$ in Eq (1) of [Uehara and Sun, 2021]). We note that if we choose $\alpha =0$, we are only considering the model with the best fit empirically, which may not be the true model. To make sure we include the true model, we need to set $\alpha$ to be a non-zero value ("large enough"), which can be informally thought of as a "variance" term that depends on sample size and the complexity of model class.

---

> > ### Comment · Reviewer_sNsB · 2023-08-15
> > **Response to the authors**
> >
> > Dear authors,
> >
> > Thank you so much for the responses, which clarify some of my previous concerns.
> >
> > I do have some follow-up questions pertaining to your responses.
> >
> >
> > 1. Regarding the statement in the global response: *"a policy cloned on the expert dataset ... is the hardest policy to show RPI against - a near-expert policy that lies outside the data support"*. I am a bit confused about why it is the "hardest" to show RPI against? And why such a near-expert policy lies outside the data support? For the latter, wouldn't "a policy cloned on the expert dataset" automatically lies in the data support?
> > 2. For *"it is straightforward to handle the case of misspecified model and policy classes by additional additive terms"*, could you briefly discuss how would you plan to handle? I understand that there is a word-limit in the response, so it is fine to only give a brief overview.
> > 3. Is the RPI shown on Figure 3 significant? Or, maybe a more fair question is: in what percentage of the settings the RPI is significant over the reference policy? I can see that on many settings, the error bars of the reference policy overlaps substantially with ARMOR (under various $\beta$). Examples may be hammer-human, door-human, hopper-med-replay, and so on.
> >     * In any case, I highly recommend the authors to re-draw Figure 3 and set the $y$-range separately for different datasets so that the distinction between REF and the ARMOR variants can be more apparent.
> > 4. *"in general it is hard to directly compare model-based and model-free methods over implementation details"* --- I agree with this. But I would like to kindly point out that both MOPO and COMBO are model-based methods. I think a more apple-to-apple comparison between ARMOR and these baselines can better demonstrate the gain of the proposed method. Otherwise, the current Table 1 looks like the improvement of ARMOR may just come from a better backbone, which ARMOR may not surpass either.
> >     * If an apple-to-apple comparison is impossible, can the authors briefly discuss what are the main obstacles (except for using dynamic ensembles which should not be a major blocker)?

---

> > > ### Author Response · Authors · 2023-08-17
> > > **Response to the Reviewer (1/2)**
> > >
> > > Thank you for the response. Please find our responses to your additional questions below:
> > >
> > > **Question 1**
> > >
> > > Regarding the statement in the global response: "a policy cloned on the expert dataset ... is the hardest policy to show RPI against - a near-expert policy that lies outside the data support".
> > >
> > > Before answering the reviewer’s question, we want to highlight that, in the RPI experiments, the data that ARMOR uses and the data that was used (by BC) to create the expert reference are **different**. In other words, ARMOR needs to learn to be competitive to the reference policy, **without** using the data that created the reference or using data running the reference would generate.
> > >
> > > **Why the expert reference is the hardest**
> > >
> > > Since the reference is near optimal, the learned policy also needs to learn to be near optimal to demonstrate the RPI property. This is highly nontrivial, because the learner doesn’t have the reference’s data (as highlighted above) and we have seen that running standard offline RL on the data the learner has access to cannot reach near optimal performance. If ARMOR achieves RPI here, it means that it must be doing something special (i.e., be effective in leveraging the reference policy). In contrast, if we were to use a reference policy with performance that is attainable by offline RL (ORL) using the training data, we wouldn’t be able to clearly establish whether the improvement over the reference is due to typical (ORL) effects, or due to RPI.
> > >
> > >
> > > **Why the near-expert policy lies outside the data support**
> > >
> > > ARMOR takes as input two things - an offline RL dataset and a reference policy. In the RPI experiments, the dataset is either the medium or medium-replay, and the reference is the “policy cloned on expert dataset”. As highlighted above, ARMOR does not have access to the actual expert dataset for training. We can further ascertain that the near-expert reference lies outside data support by looking at the performance of the best policy extracted by running ORL on medium and medium-replay datasets. This serves as a rough proxy for coverage because if the near-expert policy was covered by the dataset, ORL would recover it. Since, the best ORL policy is not near-expert we can approximately say that it lies outside data support. We realize that this is not a mathematically principled statement, but it is only meant to elucidate the overall point.
> > >
> > > **Question 2**
> > >
> > > The key idea for dealing with misspecified classes is to set the best in-class candidate as the learning goal, and pay for the gap between best in-class candidate and the ground truth (i.e., misspecification).
> > >
> > > For adapting to the model misspecification (modifying Theorem 2): Define $M_{\rm class}^\star$ to be the best in-class candidate, which has small total variation error from the ground truth $M^\star$. Then it is straightforward to get a slightly different version of Theorem 2 which bounds $J_{M_{\rm class}^\star}(\pi^\dagger) - J_{M_{\rm class}^\star}(\widehat\pi)$ rather than $J_{M^\star}(\pi^\dagger) - J_{M^\star}(\widehat\pi)$ (original version), and we pay for the difference between $2\max_{\pi} |J_{M_{\rm class}^\star}(\pi) - J_{M^\star}(\pi)|$ (i.e., misspecification, bounded by the small total variation error assumed before) to obtain what we want.
> > >
> > > Adapting to the policy misspecification (modifying Theorem 3) is similar. We define $\pi_{\sf ref}^{\rm class}$ to be the best in-class behavior estimator and then 1) using $\pi_{\sf ref}^{\rm class}$ as an intermediate comparator (i.e., $J(\pi_{\sf ref}) - J(\widehat\pi) = J(\pi_{\sf ref}) - J(\pi_{\sf ref}^{\rm class}) - [J(\widehat\pi) - J(\pi_{\sf ref}^{\rm class}])$); 2) $J(\pi_{\sf ref}) - J(\pi_{\sf ref}^{\rm class})$ bounded by policy misspecification; 3) $\pi_{\sf ref}^{\rm class}$ is in class now, we just need to follow the similar argument of dealing with model misspecification as above.

---

> > > > ### Author Response · Authors · 2023-08-17
> > > > **Response to the Reviewer (2/2)**
> > > >
> > > > **Question 3**
> > > >
> > > > First, we would like to clarify that RPI is defined as the learned policy being **not worse** than reference policy. RPI does not mean that the learned policy will be **strictly better** than the reference. For the RPI experiments in the main paper, we considered 7 environments with a total of 14 datasets (2 per environment). Overall, we observe the following metrics:
> > > >
> > > >
> > > > **In *14/14* datasets, ARMOR shows RPI (i.e., ARMOR policy is not worse than the reference policy)** (Criterion: the confidence intervals overlap) As we mentioned above, the difference between ORL and REF is a rough indication of whether the reference policy is in the data support. Here we note that in 12/14 cases REF is strictly better than ORL and in **all** those cases ARMOR demonstrates RPI.
> > > >
> > > >
> > > > **In **5/14** datasets, the ARMOR policy is *strictly* better than the reference.** (Criterion: the lower confidence of ARMOR performance is better than upper confidence of REF )
> > > > Note here that this metric is highly dependent on the quality of the reference policy. Since the reference is near-expert in these experiments, it can be hard for some environments to improve significantly over the reference.
> > > >
> > > >
> > > > Thank you for the suggestion to re-draw Fig. 3 with a different y-axis range. We will try to include this in the final draft.
> > > >
> > > > **Question 4**
> > > >
> > > > Regarding an “apples-to-apples” comparison with MOPO and COMBO, we assume the reviewer means implementing all three algorithms using a common backbone. In this regard, we would like to comment on how the conceptual differences among the algorithms led to us choosing ATAC as the backbone.
> > > >
> > > > MOPO uses SAC as the underlying RL algorithm. It modifies the reward function for SAC using a pessimism bonus based on dynamics uncertainty and trains the policy on a mixture of model-free and model-based data. In contrast, COMBO’s implementation essentially runs CQL on the mixture data. Note that in both cases the dynamics model training is **unaffected** by policy optimization.
> > > >
> > > > However, according to the theory in ARMOR, we need to train an **adversarial** model for the policy. In terms of an actor-critic framework, this means the Bellman error of the critic has to be changeable by the model (which is similar to back-propagation-through-time, but implemented via a critic). This is why the model in ARMOR also minimizes the Bellman error estimate, unlike COMBO and MOPO which train the model and policy+critic in two separate stages. In addition, Appendix D in (Cheng et al, 2022) also shows that the pessimism objective in CQL cannot facilitate adversarial training.
> > > >
> > > > Therefore, although ARMOR, COMBO and MOPO, all run actor-critic style algorithms on mixture data, we cannot implement ARMOR based on their backbones CQL or SAC (and vice versa). In fact, the changes (from CQL/SAC) required to implement the adversarial training step needed here essentially make the underlying RL backbone equivalent to ATAC. Therefore, we chose to build upon that.

---

> > > > > ### Comment · Reviewer_sNsB · 2023-08-17
> > > > > **Response to the authors**
> > > > >
> > > > > Dear authors,
> > > > >
> > > > > Thank you so much for the response, which addresses most of my concerns. I will increase my rating to 6.

---

### Official Review · Reviewer_pF8E · 2023-07-07

**Soundness:** 3 good
**Presentation:** 3 good
**Contribution:** 3 good
**Rating:** 4
**Confidence:** 3

**Summary:**

This paper introduces ARMOR (Adversarial Model for Offline Reinforcement Learning), a novel model-based framework for offline reinforcement learning (RL) that addresses the challenge of performance degradation. Offline RL allows learning decision-making policies from logged data without requiring new data collection. ARMOR robustly learns policies that improve upon a reference policy by adversarially training a Markov decision process (MDP) model. The framework optimizes policies for worst-case performance relative to the reference policy, ensuring robust policy improvement regardless of data coverage. The authors provide theoretical proofs that ARMOR competes with the best policy within data coverage and never degrades the performance of the reference policy, even when the reference policy is not covered by the dataset.

To validate their claims, the authors present a scalable implementation of ARMOR that achieves state-of-the-art performance on D4RL benchmarks without using model ensembles. This makes ARMOR suitable for using high-capacity world models. The empirical results support the robust policy improvement property of ARMOR.

**Strengths:**

Overall, ARMOR offers a promising solution for offline RL, providing robust policy improvement over a broad range of hyperparameter choices and regardless of data coverage. The framework has practical implications for real-world problems where collecting diverse or expert-quality data is expensive or infeasible. The paper contributes theoretical analysis, a scalable implementation, and empirical validation.

**Weaknesses:**

1. Inadequate Discussion of Existing Limitations: The paper briefly mentions that offline RL algorithms have not been widely adopted beyond academic research, but it fails to provide a comprehensive discussion of the existing limitations and challenges that hinder their practical adoption. Without a clear identification and discussion of the existing limitations, the paper's motivation may appear less compelling.

2. Conflicting results:  There is uncertainty regarding the claim that the policy learned by the proposed approach outperforms different hyper-parameter settings. In Figure 1, significant variations in performance can be observed for the different hyper-parameter choices. It is important to address and discuss this variation in more detail to provide a clearer understanding of the robustness and reliability of the proposed approach across different hyper-parameter settings.

**Questions:**

None

**Limitations:**

Yes

---

> ### Author Rebuttal · Authors · 2023-08-10
>
> Thank you for your time reviewing the paper and providing valuable feedback. We hope our answers below can address your concerns.
>
> **Weakness 1**: “[the paper] fails to provide a comprehensive discussion of the existing limitations and challenges that hinder [offline RL’s] practical adoption”
>
> As with any ML methodology, real-world adoption is always a complicated multi-faceted problem that comes with many challenges. For offline RL, there are many challenges, such as the difficulty of offline model selection, possible confounding in dataset, real-world non-stationarity, and the list can go on; it is virtually impossible to have a “comprehensive discussion” of all these issues in a technical paper (this is perhaps better suited for a survey/position paper).
>
> Among all these challenges, we isolate a single important challenge and address it, namely
> the risk of performance degradation from the current baseline policy. To illustrate, consider using offline RL to improve a recommendation system. The product team has a running policy (i.e., the baseline policy) right now and has some offline data, which is collected by the current running policy as well as other previous policies. Naturally, we want the policy learned by offline RL to be at least no worse than the current policy under deployment; otherwise, we should just continue running the current one. Such a scenario of learning to improve a baseline policy using mixed offline data is common in offline RL applications. Since these applications are often where making mistakes is costly (e.g., losing real money in this example), performance degradation is not allowed.
>
> The extension of robust policy improvement (RPI) to improvement over any given reference/baseline policy is exactly motivated by this. However, no existing offline RL algorithm has this guarantee to our knowledge (that is, running them with an incorrectly picked hyperparameter can produce a policy worse than the baseline policy).
>
>
> **Weakness 2**: Conflicting Results
>
> We would like to point out that the empirical results in Fig. 1 and Sec. 5.2 demonstrate that ARMOR can match or outperform the reference policy for a range of $\beta$ values, **consistent** with the theoretical results. The theoretical analysis only states that we can improve upon the reference policy for a range of $\beta$ values. It does not say the policy learned with each $\beta$ would be the same or have similar performance, just that they should all be no worse than the reference policy. In Fig. 1 and Sec 5.2, ARMOR is consistently able to outperform the reference policy for a range of $\beta$, demonstrating insensitivity to $\beta$. Further, we would like to point the reviewer to our global response and the uploaded PDF,  where we provide further empirical evidence to support that the RPI property holds for different reference policies as well.

---

> > ### Comment · Reviewer_pF8E · 2023-08-21
> >
> > Thank you for the response. I still don't find the reasoning for the weaknesses convincing, and will keep the score of 4.

---

### Official Review · Reviewer_suos · 2023-07-08

**Soundness:** 3 good
**Presentation:** 3 good
**Contribution:** 3 good
**Rating:** 7
**Confidence:** 4

**Summary:**

The paper propose a new model-based offline RL framework, called ARMOR, which can robustly learn policies that improve upon an reference policy by adversarially training a Markov decision process (MDP) model.  ARMOR aims to optimize for the worst-case relative performance over uncertainty. In experiment, ARMOR implementation achieves good performance on D4RL benchmarks using only a single model.

**Strengths:**

1. The idea is novel. The paper give us a new perspective to learn a dynamics model in offline RL, which is valuable to the community;
2. The paper is overall well-written and easy to follow for me;
3. The implementation is generally reasonble in intuition.

**Weaknesses:**

1. The description in Section 4.1 is quite confusing, with many details tucked away in the appendix. This arrangement is not particularly reader-friendly. I suggest that the authors better link the following points: (1) why the objective in Line 228 can substitute for Eq. (1); (2) how Line 228 is transformed into the goal of Line 199 and Algorithm 1. You don't need to elaborate on the relationship, but at least some intuition or motivation should be provided to the reader. Additionally, an optional suggestion: the authors might consider restructuring the overall arrangement by first introducing the part of Line 228 and then describing how it is implemented, which may result in a more natural discourse.

2. The notation used for expressing errors is confusing here, with many instances of $\mathcal{E}$ and $\mathcal{L}$ representing loss, lacking in distinctiveness. A classic example is the symbols in Line 228, $\mathcal{E}\_{D}$  and $\mathcal{E}_{\hat D}$, both error symbols share the same structure, only the parameters differ. But they represent completely different losses, with the former being the maximum likelihood, and the latter the Bellman error. I suggest the authors revise the notations used, making it clear to readers what kind of loss it is without having to inspect the parameter differences.


3. Figure 2 could be further polished: Without reading the main text, readers would not understand why ARMOR chooses the two models on the right, as the figure (including the caption) does not demonstrate that the reference policy is actually moving rightward. I believe the authors could add this information, allowing readers to better grasp the authors' motivation.

3. Although D4RL is a fairly popular benchmark, I think the tasks from D4RL do not adequately highlight the advantages of the proposed algorithm. As the authors mentioned, this algorithm is motivated by real-world application scenarios, like "Usually, the systems we apply RL to have currently running policies, such as an engneeded autonomous driving rule or a heuristic-based system for diagnosis, and the goal of applying a learning algorithm is often to further improve upon these baseline reference policies...". However, the data in the selected D4RL datasets mostly come from mixed policies. The medium dataset aligns with this scenario, but the baselines already perform well, making it hard to discern the advantage of the proposed method. The recently released neorl [1] dataset might be more suitable for this work, as it was specifically designed for the scenarios the authors proposed, with all datasets collected by a single working policy, a concept similar to the authors' reference policy.



[1] NeoRL: A Near Real-World Benchmark for Offline Reinforcement Learning

**Questions:**

The implementation of Equation (4) somewhat contradicts my intuition. Specifically,
- $M$'s initial goal was to enhance $\pi_{\rm ref}$'s $J$ and reduce $\pi$'s $J$. However, in the final implementation, none of $M$'s optimization aspects involve $\pi_{\rm ref}$. This makes it unclear how the losses employed in practice can meet the original optimization objective of Equation (1).
- I'm unclear on why the optimization process for the critic necessitates a pessimistic loss, e.g., $\mathcal{L}_{D_M}$? The need for pessimism about the critic's estimates isn't evident in the objective Equation (1).

In Table 1, I'm curious on why several cloned datasets don't work by ARMOR? This seems to be the datasets very suitable for ARMOR to show better performance.

In Figure 3, the algorithm shows sensitivity to $\beta$. It would be beneficial if the authors could elucidate the underlying reasons for this and provide any principles to guide researchers.

The experiments in Section 5.2 could be improved. The authors claim that "ARMOR can robustly learn policies that improve upon an arbitrary reference policy..." which is important. However, in the experiments, only performance improvements under a specific reference policy are demonstrated. It would be more constructive to verify this claim by picking various policies as reference policies and conduct experiments under one or two environments. I think DOPE [1] can be referenced as a work to conduct this experiment, which preserves many policies that can be used as reference policies.

Regarding related work, a discussion could be made with the work [2] that utilizes adversarial approaches for offline environment learning. The optimization objective of this work coincidently using an opposite objective to the Equation (1) of this paper for model learning, which is "$\max_M \min_\pi \ell(M, \pi)$".

Also suddenly inspired by the work [2], I would like to propose an open question for discussion: Can we also consider using adversarial policies further with ARMOR, such as $\max_\pi \min_M \max_{\pi_{\rm ref}}$? Since constructing $\pi_{\rm ref}$ from the offline data isn't a necessity, following Theorem 3, if we can create a sufficient number of $\pi_{\rm ref}$ in an adversarial manner, could we theoretically ensure that the final $\hat \pi$ approximates the optimal policy in any situation?

[1]Benchmarks for Deep Off-Policy Evaluation.

[2] Adversarial Counterfactual Environment Model Learning.

**Limitations:**

NAN

---

> ### Author Rebuttal · Authors · 2023-08-10
>
> Thank you for your detailed feedback on improving the readability and constructive comments. We hope that our responses below would help resolve the remaining concerns you might have.
>
> **W 1**: We thank the reviewer for pointing this out.
>
> [How Eq.(1) → L228]:
> We have a detailed discussion regarding this in Appendix C due to the space limit. The high-level overview is: (1) Relax constrained optimization to regularized optimization. (2) Utilize the learned model to generate augmented data to perform relative pessimism properly (this is why ARMOR could have stronger RPI than the model-free opponents, e.g., ATAC).
>
> [How L228 → L199 and Algorithm 1]:
> We would like to clarify that the goal of Algorithm 1 is essentially optimizing L229, whereas L199 denotes how each term on L228 is estimated in Algorithm 1.
>
> **W 2**: Currently, we use $\mathcal{L}$ to denote the term that the policy and the model compete with, and $\mathcal{E}$ to denote all other losses for enforcing data-consistency on the model. Different $\mathcal{E}$ losses have different function signatures and we always include the input parameters the loss takes to make it clear. We will make them more visually different, such as by adding additional superscripts.
>
> **W 3**: Please refer to the Global Response and the updated figure in our uploaded PDF.
>
> **W 4**: Thanks for pointing this out. NeoRL seems like a great suite to test our algorithm in the future. However, it is relatively new and getting infrastructure setup to run experiments along with baselines takes a non-trivial amount of time. Also please note that our experiments are not limited to classic D4RL experiments. Here we have also constructed modified problems from D4RL to test RPI (sec 5.2 and Fig. 3): We specifically chose an expert reference policy that is beyond what the data can cover to showcase the benefit of RPI. In almost all cases, ARMOR using the expert reference can outperform the offline RL baseline and sometimes even the expert reference.
>
> **Q 1**: The model and pessimistic losses are connected to each other; as a result, the model is learned to maximize the performance difference between the two policies. This is implemented in Alg. 1 by *jointly* optimizing the critic and the model to be pessimistic. We let both the critic and the model minimize the Bellman error (the critic: pessimistic loss+ *Bellman error*; the model: *Bellman error*+ model fitting loss). So when the critic minimizes the pessimistic loss (to maximize the performance difference), the model also becomes pessimistic due to the link of Bellman error. We explained this in Lines 210-216 of the main paper, where we discuss how setting the weight on model fitting loss to be zero makes ARMOR equivalent to Imitation Learning (this is also verified empirically in Appendix G). We chose to implement ARMOR this way since it is more computationally efficient than directly optimizing Eq.(1) which would require back-propagation through full length rollouts generated by the model for every policy update.
>
> **Q Table 1**: The performance on the cloned datasets is on-par with the baselines. While in some cloned datasets ARMOR is not the best, no algorithm gives meaningful scores there (please see the scale in the -exp version). Since we implement the approximate version with neural networks, it is hard to always ensure perfect consistency between the theory and empirical results.
>
> **Q Fig 3**: We would like to point out that the results in Fig. 1 and Sec. 5.2 in fact show insensitivity to hyper-parameters for the small $\beta$ regime, *consistent* with the theory. The theory states that we can improve upon the reference policy for a range of beta values. It does not assert that the learned policy with each beta would be the same, or have similar performance, just that they should all be no worse than the reference policy. In Fig 1, ARMOR is consistently able to outperform the reference policy for a range of beta, demonstrating insensitivity to $\beta$.
>
> **Q Sec. 5.2**: Due to space constraints, we chose the hardest reference to compete with, i.e., a near-expert policy that lies outside the data support. This is because we know that running standard offline RL cannot achieve the expert level performance; as a result, if ARMOR achieves the expert performance, it must be due to RPI. Based on the review, we have conducted more experiments to show that ARMOR can achieve RPI with respect to arbitrary policies for a very wide range of $\beta$ values. Please refer to the Global Response for more details.
>
> **Q Related Work [2]**: [2] considers a worst-case model fitting error (their Eq (3)), $ \min_M \max_{d^{\pi_\beta}} [ l_M( s,a,s’) ]$, where $l_M(s,a,s’)$ is the model fitting loss (e.g., the term in the sum of our Eq (3)), and then use the learned model for policy optimization. In contrast, ARMOR finds an adversarial model for each policy, and optimizes the policy that performs well under its adversarial model. We highlight that when the offline data does not have full coverage, their approach could lead to a poorly performing policy, because it does not consider the quality of model predictions outside data support, where the model can be arbitrarily bad. This makes their method inapplicable to offline RL under partial coverage setting here.
>
> **Q Open question**: We think your proposal is quite interesting. We suppose you meant modifying our objective in Eq. 1 to $ \hat{\pi} = \arg\max_{\pi} \min_M  \max_{\pi_{ref}} J_M(\pi) - J_M(\pi_{ref})$. We actually have a long, dedicated section on this in Appendix E (see line 641) and we call this approach Regret Minimization (as it tries to find the policy that has the smallest worst-case regret). Its worst case regret can be analyzed in a similar fashion as ARMOR. However, this algorithm can be overly conservative and does not necessarily have RPI to a given reference policy. We can discuss more in the final draft due to space limitation in rebuttal.

---

> > ### Comment · Reviewer_suos · 2023-08-18
> >
> > Thank you for your response. Most of my concerns have been addressed.
> >
> > Regarding the open questions:
> >
> > 1. I reviewed Appendix E, but I'm still struggling to understand the relationship between $\max_\pi \min_M \max_{\pi_{ref}} J_M(\pi) - J_M(\pi_{ref})$ and Regret Minimization. How does this objective potentially lead to an overly conservative policy?
> >
> > 2. Assuming that $\max_\pi \min_M \max_{\pi_{ref}} J_M(\pi) - J_M(\pi_{ref})$ results in an overly conservative policy, I'm trying to reconcile this with my understanding of Theorem 3. Specifically, if we can generate a sufficient number of $\pi_{ref}$ adversarially, wouldn't we theoretically ensure that the final policy approximates the optimal policy in any situation? This appears to be a direct implication of your Theorem 3. Have I overlooked something?

---

> > > ### Author Response · Authors · 2023-08-18
> > >
> > > Thanks for your further question.
> > >
> > > First apologies. We made a typo. What we intended to mean in the rebuttal was $\arg\max\_\pi \min\_M {\color{red} \min\_{\pi\_{ref}}} J\_M(\pi) - J\_M(\pi\_{ref}) = \arg\max\_\pi \min\_M  J\_M(\pi) -  \max\_{\pi\_{ref}}  J\_M(\pi\_{ref})$, otherwise the innermost optimization would try to pick the easiest reference policy (i.e., the worst policy in $M$) to compare with and is no longer adversarial. We hope that this is the same as what the reviewer meant earlier.
> > >
> > > After this correction, suppose the innermost $ \min\_{\pi\_{ref}}$ is over a sufficiently rich policy class, then it's easy to see that the minimizer is $\pi\_{ref} = \pi\_M^\star$, that is,
> > > $\max\_\pi \min\_M { \min\_{\pi\_{ref}}} J\_M(\pi) - J\_M(\pi\_{ref})
> > > = \max\_\pi \min\_M J\_M(\pi) - J\_M(\pi\_M^\star)$, thus reducing to the regret minimization case discussed in Corollary 10, bullet point 3. We said the “this algorithm can be overly conservative” because the term $J\_M(\pi) - J\_M(\pi\_M^\star)$ above is always greater than or equal to the performance difference for any fixed reference policy $\pi_{ref}$, i.e., $J\_M(\pi) - J\_M(\pi\_{ref})$. As a result, the policy learned in this way does not necessarily have RPI to a given $\pi_{ref}$.
> > >
> > > And to reconcile with Theorem 3, note that Theorem 3 applies to ARMOR which has **a fixed $\pi\_{ref}$**, but the $\pi\_{ref}$ above corresponds to a policy that changes as we consider different models $M$ in the version space. When $\pi\_{ref}$ is fixed, it is easy to show (under the assumptions we make) that the objective value of ARMOR is never negative (i.e., RPI), so we never have degenerate performance. The problem with regret minimization and using multiple $\pi\_{ref}$ in general is that the objective value can become negative, which is a sign that the optimization is ill-posed and we are being overly conservative about the performance difference to $\pi_{ref}$.
> > >
> > > Related to this point, the reviewer conjectured that "[with] sufficient number of $\pi\_{ref}$, [we can] theoretically ensure that the final policy approximates the optimal policy". This is not true. Here is a minimal counterexample:
> > >
> > > Consider a 2-armed multi-armed bandit. Suppose our version space is $(1, 0), (0, 1)$, i.e., we know one arm is good (reward 1) and one arm is bad (reward 0), but we don't know which is which. In this case, if we choose the two deterministic policies as the 2 reference policies and plug them into the formulation above, the optimal objective is $-1/2$ and the best policy is choosing between the two arms uniformly randomly. Such a policy is suboptimal in both model instances, and in general no policy can yield good improvement in such a case. In general, being able to “approximate the optimal policy” without further assumptions is too good to be true in offline RL since there is model uncertainty that we cannot resolve without access to further (online) data.

---

> > > > ### Comment · Reviewer_suos · 2023-08-19
> > > >
> > > > Thanks for the response.  The meaningful adversarial objective should be $\max_\pi \min_\pi \min_{\pi_{ref}} J_M(\pi) - J_M(\pi_{ref})$. Sorry for my mistake. After reading the response, I have recognized that the objective will lead to an overly conservative policy based on the intuitive example. But after recapping the assumptions in the main body, I still cannot understand the statement you made "It is easy to show (under the assumptions we make)  ... conservative about the performance difference to $\pi_{ref}$". How can the assumptions (Assumption 1 and Assumption 2) you make derive the conclusion that "the objective value of ARMOR is never negative" and how does the negative value make the optimization ill-posed?

---

> > > > > ### Author Response · Authors · 2023-08-19
> > > > >
> > > > > > "The objective value of ARMOR is never negative"
> > > > >
> > > > > For this we only need the realizability of $\pi\_{ref}$ in Assumption 2, that $\pi\_{ref} \in \Pi$. This way, if we simply choose $\pi = \pi\_{ref}$, the objective is $0$ ($J\_M(\pi) - J\_M(\pi\_{ref})$ is always $0$ regardless of $M$), and since we maximize $\pi$ the optimal choice must have an objective no lower than that of $\pi = \pi\_{ref}$.
> > > > >
> > > > > > "how does the negative value make the optimization ill-posed"
> > > > >
> > > > > This is not a strict statement but more of an intuition. Roughly speaking this means that the improvement goal is too hard to achieve, and in the worst case not only we cannot guarantee improvement, but our policy may be significantly worse than the reference policy. We recommend going through the 2-armed bandit example we provided in the previous response for concreteness.

---

> > > > > > ### Comment · Reviewer_suos · 2023-08-19
> > > > > >
> > > > > > Thanks for the response. If so, I think something is missing in the theorem since, as you said, "the negative value makes the optimization ill-posed" is just from an intuition, instead of a necessity in the process of theoretical analysis. Thus, if I understand correctly, we can derive a contradictory conclusion from intuition and your theorem?
> > > > > >
> > > > > >
> > > > > > In addition, in my previous review, my concerns are major in the written issue, thus I recommend the authors give a specific revision plan (with Line numbers or Section/paragraph). I am glad to increase my score if the written issue is addressed well and the above concern is solved.

---

> > > > > > > ### Author Response · Authors · 2023-08-19
> > > > > > >
> > > > > > > From reading the reviewer’s response, we found that there is likely a confusion, which might have been due to this thread touching on multiple topics. Please let us clarify.
> > > > > > >
> > > > > > >
> > > > > > > > something is missing in the theorem
> > > > > > >
> > > > > > > We would like to emphasize there is no contradiction between our theorem and the discussions above. We clarify the following logical steps:
> > > > > > >
> > > > > > > 1. The theorems in the paper are for ARMOR, which takes a **fixed** $\pi_{ref}$ as input to the algorithm.
> > > > > > > 2. Under Assumption 2, the optimal objective of ARMOR is non-negative (as we explained in the previous response).
> > > > > > > 3. The proof of Theorem 3 on RPI **crucially** relies on this non-negative property (see line 500 in Appendix C).
> > > > > > >
> > > > > > > Our current analysis is **not** applicable to other algorithms such using multiple reference policies, or adapting the reference policy during learning, as we discussed in this thread of **open** questions. For these non-ARMOR algorithms, the optimal objective value can be negative (as we explained in the previous response), which breaks the proof of Theorem 3 about RPI (Step 3 above). Thus, we cannot show these non-ARMOR algorithms have RPI to a given $\pi_{ref}$ using the current analysis. Further, we believe these non-ARMOR algorithms don’t have RPI in general, because we can construct counterexamples (as we demonstrated using the bandit example in the previous response).
> > > > > > >
> > > > > > > We didn’t give a precise definition of what “ill-posed” means in our first response, which is why we said it’s “more of an intuition”. But we highlight that all the statements in our paper are formally proved. In addition, we point out above non-negativity is crucial to establish the RPI property. So by “ill-posed” what we meant more precisely is that it breaks the current proof and therefore we cannot establish performance guarantees for these non-ARMOR algorithms in the same way we did for ARMOR. We hope this discussion would resolve the reviewer’s remaining concerns.
> > > > > > >
> > > > > > > > revision plan
> > > > > > >
> > > > > > > We thank the reviewer for suggestions on improving the writing clarity. We will implement them according to our initial rebuttal (e.g., about sec 4.1, loss notation, figures, etc.). We also think this line of “open-question” discussion can be informative and helpful to future readers. We will incorporate it to improve the paper’s clarity (such as highlighting that  ARMOR takes a fixed reference policy, and why non-negativity resulting from Assumption 2 is important to establish RPI). We also welcome any further suggestions from the reviewer.

---

> > > > > > > > ### Comment · Reviewer_suos · 2023-08-20
> > > > > > > >
> > > > > > > > If the non-negative property is a necessity in the process of theoretical analysis, it is ok for me as the theorem is consistent with the intuition. Could you check the line you mentioned to show the non-negative property, since I found line 500 is at the end of Appendix A.4 instead of Appendix C, and it is an uninformative sentence "This completes the proof"?

---

> > > > > > > > > ### Author Response · Authors · 2023-08-20
> > > > > > > > >
> > > > > > > > > Line 500 is correct. But we made a typo; you're right it's Appendix A.4, not Appendix C. Sorry for the confusion.
> > > > > > > > >
> > > > > > > > > What we referred to is the proof just below line 500, which shows "Proof of Theorem 3"; recall Theorem 3 is the RPI theorem of ARMOR. The proof uses the non-negativity of the optimal objective value: $ \max_{\pi \in \Pi} \min_{M \in \mathcal{M}} [ J_M(\pi) - J_M(\pi_{ref})] \geq \min_{M \in \mathcal{M}} [ J_M(\pi_{ref}) - J_M(\pi_{ref})]  = 0$, where the inequalities uses $\pi_{ref} \in \Pi$ which is Assumption 2. (This is the last two lines of the proof below line 500). We hope this is clear now!

---

> > > > > > > > > > ### Comment · Reviewer_suos · 2023-08-20
> > > > > > > > > >
> > > > > > > > > > Everything is clear now :)
> > > > > > > > > >
> > > > > > > > > > About the revision plan:
> > > > > > > > > > 1. I think Section 4.1 can be restructured to improve readability. Currently, readers might need to repeatedly check all of the  terms among Equations in Line 199, Eq (4-5), and the objective in Line 228 to sort out their relationship.  After reading the rebuttal, in my opinion, we can place the "Connection to the Theoretical Formulation" first, which transits the story from Eq (1) to the real implementation and give the whole and real objective, then give the description and insights of terms (like Line 199), and finally give the whole algorithm and other implementation details.
> > > > > > > > > > 2. I would like to see more discussion on related studies that use the adversarial way for model learning. At least the study [2] is worth to be discussed as a related work since the study also considers an objective of adversarial model learning, in an opposite way.
> > > > > > > > > >
> > > > > > > > > > Thanks for your detailed response. I will increase my score.

---

### Official Review · Reviewer_V62y · 2023-07-12

**Soundness:** 3 good
**Presentation:** 3 good
**Contribution:** 3 good
**Rating:** 6
**Confidence:** 3

**Summary:**

The paper introduces the Adversarial Model for Offline Reinforcement Learning (ARMOR), a model-based offline RL framework. ARMOR uses adversarial training to robustly learn and improve policies over any given reference policy, regardless of data quality. The framework utilizes the concept of 'relative pessimism' for worst-case optimization, ensuring that it either maintains or improves upon the performance of the reference policy, a property known as Robust Policy Improvement (RPI). The authors have also shown, theoretically and empirically, that ARMOR is robust to hyperparameter choices, and can outperform or be on par with state-of-the-art offline RL methods.

**Strengths:**

* The paper is well-written and technically sound. The RPI in this paper is stronger than the RPI property in the literature, which only guarantees to be no worse than the behavior policy that collected the data.
* The experimental results show the effectiveness of the proposed method and it outperforms existing baselines in many environments.


**Weaknesses:**

* I think the proposed method highly depends on the quality of the MDP model $M$. For the results in Table 1, in some environments, the proposed method does not outperform baselines. I think some explanation and intuitions are needed. I am wondering if the $M$'s structure is not optimal in those cases.
* The model fitting loss in Equation (3) is not well-defined. Specifically, $r$ is not mentioned. Is it the same as $R^*$ as in Definition 1?
* Minor: Please refrain from only using color to distinguish bars in Figure 3, as it is not friendly to readers with color blindness. Also, in Figure 2, not all illustrations are valid MDPs. I suggest the authors re-draw those toy examples.

**Questions:**

Please refer to the weakness section.

**Limitations:**

The authors adequately addressed the limitations.

---

> ### Author Rebuttal · Authors · 2023-08-10
>
> Thank you for your constructive feedback. We will incorporate them in the revision. We have addressed your comments below.
>
> **Weakness 1**
>
> We would like to clarify that ARMOR (conceptually) maintains a set of models (i.e., the version space), not just a single one. We also assume that the model class is rich enough in theory such that it includes the true model $M^*$ (Assumption 1, realizability). Note that ARMOR doesn't need to explicitly recover the true $M^*$;  by maintaining a model set that contains $M^*$ and performing worst-case optimization over the set, it automatically enjoys the robustness guarantees (i.e., not being worse than the reference policy). However, robustness here comes at a cost, as worst-case reasoning over *a set of models* may lead to a conservative policy that does not aggressively optimize the return. Hence, there is a fundamental robustness vs. performance trade-off. In Table 1, we see ARMOR is comparable to other baselines across most datasets, with the only exception being halfcheetah domains when compared with RAMBO (in halfcheetah, ARMOR still performs better than other non-RAMBO baselines).
>
> **Weakness 2**
>
> In Eq (3), $R_M$ denotes the predictions made by the model $M$ and $r$ denotes the reward labels in the data, as per the definitions in Sec. 2. We will make this clearer in the final draft.
>
> **Weakness 3**
>
> Fig. 3: Thank you for pointing this out. We will update in the final draft.
>
> Fig. 2: We are unsure what the reviewer means by “in Figure 2, not all illustrations are valid MDPs.” It would be helpful if the reviewer could elaborate on this so we can make the appropriate changes if required.

---

### Author Rebuttal · Authors · 2023-08-10

We would like to thank all the reviewers for taking the time to review our work and providing their constructive feedback. Here we provide responses to some of the common concerns


**RPI Experiments**

Reviewers “suos”, “pF8E” and “sNsB” had questions about the RPI experiments, specifically regarding sensitivity to hyper-parameter values and the nature of the reference policy. We believe that the primary source of confusion could be a misunderstanding of the theoretical results and the main claims of the paper.

First, the theoretical analysis in Sec 3.2 and discussion in Sec. 6 state that ARMOR can improve upon the reference policy for a range of values for only the pessimism hyper-parameter $\beta$. They do not assert that the policy learned with each $\beta$ would be the same or have similar performance, just that they should all be no worse than the reference policy. Our results in Sec 5.2 and Fig. 1 clearly demonstrate this.

Second, we chose a policy cloned on the expert dataset as the reference in Sec 5.2 because of space limitations and the fact that it is the hardest policy to show RPI against - a near-expert policy that lies outside the data support. Based on reviewer feedback, we have conducted further experiments to demonstrate that RPI holds for other reference policies as well. In our uploaded PDF, we provide empirical evidence for RPI against a policy cloned on the random dataset as the reference, as well as a hand-designed bang-bang control policy for an even wider range of $\beta$ values. These new experiments further bolster the point that ARMOR can obtain RPI with respect to arbitrary reference policies (which might be out of support) and is robust to the pessimism hyper-parameter $\beta$.


**Illustrative Example Figure**

Reviewer “suos”  pointed out the lack of clarity about the reference policy in Figure 2 and that the caption was not self contained. We have followed up on this input and have provided an updated figure and caption as part of our response PDF, to clearly denote what the reference policy is (changes in red). We also point the reviewers to a more comprehensive example in the appendix where we show that the same holds even when the reward function is learned.



We hope that our responses and new empirical results have alleviated the concerns the reviewers raised and they will consider updating their ratings accordingly.

---

### Decision · Program_Chairs · 2023-09-21

**Decision:**

Accept (poster)

**Comment:**

This paper addresses offline Reinforcement Learning by optimizing policies for the worst-case performance relative to the reference policy through adversarially training a Markov decision process model. It is theoretically shown to compete with the best policy within data coverage when the reference policy is supported by the data. With a scalable implementation, ARMOR is shown competitive on D4RL benchmarks using a single model.

The paper addresses an interesting problem, and the method is novel.  While the reviewers had a few concerns such as sensitivity to hyper-parameters in RPI experiments, extensive back-and-forth discussion between authors and reviewers finally brings the paper to the accepted side.  Please make sure to incorporate into the revised paper the clarifications in the rebuttal.